# Inert and seed-competent tau monomers suggest structural origins of aggregation

Hilda Mirbaha[1], Dailu Chen[1], Olga A Morazova[2], Kiersten M Ruff[3], Apurwa M Sharma[1], Xiaohua Liu[4], Mohammad Goodarzi[4], Rohit V Pappu[3], David W Colby[2], Hamid Mirzaei[4], Lukasz A Joachimiak[1]*, Marc I Diamond[1]*

[1]Center for Alzheimer's and Neurodegenerative Diseases, University of Texas Southwestern Medical Center, Dallas, United States; [2]Department of Chemical and Biomolecular Engineering, University of Delaware, Newark, United States; [3]Department of Biomedical Engineering, Washington University in St. Louis, St. Louis, United States; [4]Department of Biochemistry, University of Texas Southwestern Medical Center, Dallas, United States

*For correspondence:
lukasz.joachimiak@ utsouthwestern.edu (LAJ); Marc.Diamond@UTSouthwestern. edu (MID)

Competing interests: The authors declare that no competing interests exist.

**Abstract** Tauopathies feature progressive accumulation of tau amyloids. Pathology may begin when these amplify from a protein template, or seed, whose structure is unknown. We have purified and characterized distinct forms of tau monomer—inert ($M_i$) and seed-competent ($M_s$). Recombinant $M_s$ triggered intracellular tau aggregation, induced tau fibrillization in vitro, and self-assembled. $M_s$ from Alzheimer's disease also seeded aggregation and self-assembled in vitro to form seed-competent multimers. We used crosslinking with mass spectrometry to probe structural differences in $M_i$ vs. $M_s$. Crosslinks informed models of local peptide structure within the repeat domain which suggest relative inaccessibility of residues that drive aggregation (VQIINK/VQIVYK) in $M_i$, and exposure in $M_s$. Limited proteolysis supported this idea. Although tau monomer has been considered to be natively unstructured, our findings belie this assumption and suggest that initiation of pathological aggregation could begin with conversion of tau monomer from an inert to a seed-competent form.

DOI: https://doi.org/10.7554/eLife.36584.001

## Introduction

Amyloids are ordered protein assemblies, typically rich in beta sheet, that underlie multiple disorders such as Alzheimer's disease (AD). Amyloid-forming proteins include tau, synuclein, and expanded polyglutamine proteins such as huntingtin, among many others. It is unknown how or why intracellular proteins such as tau transition from a relatively inert form to one that efficiently self-assembles into ordered structures in vivo. This process begins with the formation of a pathogenic 'seed,' a structure that serves as a template for homotypic fibril growth. This structural transition could be a critical event in the pathogenesis of neurodegeneration. Under defined conditions and relatively high concentrations (typically micromolar), recombinant tau monomer will form amyloid fibrils in vitro. However the basis of spontaneous assembly in cells is unknown. The conversion of a protein from a monomer to a large, ordered multimer could occur by several mechanisms, but the first step probably involves the formation of a seed. This event, and indeed the actual conformation or assembly state of the protein that constitutes the 'minimal' seed, has remained obscure. This has led to the idea that a seed is potentially transitory, arising from an equilibrium between two states: one relatively aggregation-resistant, and another that is short-lived. A seed could be a single molecule, or several. Based on extrapolation from kinetic aggregation studies, it has been suggested that a critical seed for tau and polyglutamine peptide amyloid formation is a single molecule (*Chirita et al., 2005*; *Bhattacharyya et al., 2005*; *Kar et al., 2011*), while an earlier study (among others

**eLife digest** When doctors perform autopsies to look at the brain tissue of people with Alzheimer's disease, they find toxic buildups of certain proteins – in particular, a protein called tau – in structures called 'aggregates'. People with more severe dementia have more tau aggregates in their brain. Aggregates form when individual proteins stick together in repetitive patterns, much like the way a single Lego block might attach to another identical one. Like all proteins, tau is built from a string of amino acids that folds into a specific shape. Normally folded tau proteins do not form aggregates. It was not clear that an individual tau protein had two distinct forms—structures associated with health ("good") or disease ("bad").

Mirbaha et al. have now studied the folding pattern of purified tau proteins with a sophisticated technology called mass spectrometry. This technique can measure changes in tiny amounts of protein. Some of the purified proteins had been extracted from human brains (from people with and without Alzheimer's). To detect which of the proteins were toxic, Mirbaha et al. also grew simple human cells in a dish that were designed to react specifically to the bad forms of tau. This allowed the good and bad forms of tau to be isolated.

Mirbaha et al. discovered that in the good form of tau the parts of the protein that allow it to stick to itself are hidden, folded inside. By contrast, the bad form of tau exposes the parts that allow it to aggregate, enabling the protein to build upon itself to form a large, toxic assembly.

The shape-shifting concept established by Mirbaha et al. might apply to other proteins that form toxic aggregates. This could help us to better understand how many other neurodegenerative diseases develop and progress. Recognizing that the shapes that tau forms can be categorized as either 'good' or 'bad' may also help to develop new treatments for Alzheimer's disease. Drugs could be designed to stabilize the good form of tau, or to help remove the bad form from the brain. Furthermore, if the shape-shift described by Mirbaha et al. can be measured early enough in patients, it may allow treatments for Alzheimer's before people have developed any detectable symptoms.

DOI: https://doi.org/10.7554/eLife.36584.002

[*Ramachandran and Udgaonkar, 2013*]) has proposed a tau multimer (*Friedhoff et al., 1998*). Isolation of the seed-competent form of tau could be critical to understanding the initiation of disease and the design of more effective diagnostics and therapeutics.

Tau forms amyloids that underlie neurodegeneration in a variety of neuropathological syndromes, collectively termed tauopathies (*Lee et al., 2001*). These include AD and frontotemporal dementias, among many others. Multiple groups, including ours, have now observed that tau will propagate an aggregated state from the outside to the inside of a cell, between cells, across synapses, and within brain networks (*Sanders et al., 2016*). In prior work, we used size exclusion chromatography (SEC) to define tau trimers as the minimal unit of spontaneous cellular uptake and intracellular amyloid formation, and proposed this as the smallest particle capable of propagating aggregates between cells (*Mirbaha et al., 2015*). This work involved application of 'naked' protein assemblies derived from recombinant protein or human brain onto cultured 'biosensor' HEK293 cells or primary neurons that express a tau aggregation reporter (*Frost et al., 2009a*; *Holmes et al., 2014*). Biosensor cells and primary neurons alike take up tau aggregates via macropinocytosis (*Holmes et al., 2013*). The aggregates subsequently serve as highly specific templates to trigger intracellular amyloid formation (*Holmes et al., 2014*; *Sanders et al., 2014*). We have also determined that preincubation of cationic lipids such as lipofectamine with tau seeds facilitates their direct transduction into a cell, bypassing the physiologic uptake mechanism (*Holmes et al., 2014*; *Furman et al., 2015*). Lipofectamine-mediated delivery into biosensor cells allows direct quantitation of seed titer for both tau and α-synuclein (*Holmes et al., 2013*).

Tau is intrinsically disordered upon isolation from bacteria or mammalian cells and is relatively inert in terms of spontaneous self-assembly. However under various conditions, including exposure to polyanions such as heparin, tau will form aggregates via nucleated self-assembly (*Goedert et al., 1996*; *Pérez et al., 1996*). It is unknown how these experimental conditions relate to the initiation of aggregation in human brain. We have now purified various stable forms of full-length tau monomer

from recombinant sources and human brain. One is relatively inert and is stable for long periods. Another is 'seed-competent,' triggers amyloid formation in cells and in vitro, and exhibits intrinsic properties of self-assembly. We have used crosslinking with mass spectrometry (XL-MS) to probe the structures of these molecules. Models of discrete regions within the RD predict that differential exposure of hexapeptide motifs previously known to be important for amyloid formation distinguishes the two forms of tau. These models are supported by limited proteolysis studies. The identification of distinct and stable forms of tau monomer, including some that are uniquely seed-competent, bears directly on how we understand the initiation of protein aggregation in the tauopathies.

## Results

### Isolation of fibril-derived monomer and other assemblies

We initially sought to define the tau seeding unit that would trigger intracellular aggregation upon direct delivery to the cell interior. We had previously observed that a tau trimer is the minimal assembly size that triggers endocytosis and intracellular seeding (*Mirbaha et al., 2015*). These experiments depended on spontaneous cell uptake, since no lipofectamine was added to the reactions. A prior study had also indicated the role of disulfide linkages in promoting tau aggregation, potentially by dimer formation (*Friedhoff et al., 1998*). Thus, for our initial studies we engineered and purified full-length (FL) tau monomer that lacks any internal cysteines due to alanine substitutions (C299A and C322A), termed tau (2A). FL tau (2A) cannot self-associate based on disulfide linkages, which helped prevent the formation of cryptic dimers that could have confounded our studies. These substitutions did not affect tau purification, heparin-induced fibrillization, and sonication protocols, which we performed as described previously (*Mirbaha et al., 2015*). We treated fibril preps with sonication, prior to isolation of recombinant FL tau (2A) assemblies of various sizes by size exclusion chromatography (SEC) (*Mirbaha et al., 2015*). In parallel, we also studied FL wild type (WT) tau. We purified unfibrillized recombinant FL tau (2A) monomer by SEC (*Figure 1A*), and isolated SEC fractions of sonicated fibrils that contained putative monomer, dimer, trimer and ~10 mer (*Figure 1B*).

### Fibril-derived monomer exhibits seeding activity in cells and in vitro

To test the seeding activity of the tau preparations, we used a previously described 'biosensor' cell reporter line (*Holmes et al., 2014*). These cells stably express 4R tau repeat domain (RD) containing the disease-associated P301S mutation. All cells express 4R-RD-Cyan fluorescent protein and 4R-RD-yellow fluorescent protein (RD-CFP/YFP). Exogenously applied seeds induce intracellular aggregation with resultant fluorescence resonance energy transfer (FRET) between CFP and YFP that can be measured via flow cytometry (*Holmes et al., 2014*; *Furman et al., 2015*). The degree of aggregation is scored using 'integrated FRET density' (IFD), which is the product of the percent positive cells and the mean fluorescence intensity of FRET-positive cells, and from this we determine a titer of tau seeding activity (*Holmes et al., 2014*). Lipofectamine directly transduces tau assemblies across the plasma membrane and increases the assay's sensitivity by approximately 100-fold. Upon incubation with Lipofectamine, we were surprised to observe seeding by monomer and larger assemblies alike, whether FL WT or 2A. (*Figure 1C,D*). Epifluorescence microscopy confirmed the presence of intracellular inclusions after FL WT tau monomer seeding (*Figure 1D*). We termed the inert monomer '$M_i$,' and the seed-competent monomer '$M_s$.' To rule out higher order assemblies of tau within the putative monomer fraction, immediately prior to the seeding assay we passed fractions through a 100 kDa cutoff filter to eliminate anything larger than a monomer. While monomer fraction retained ~80% of seeding activity, only ~20% of dimer seeding activity remained, and ~1–2% of trimer seeding activity remained (*Figure 1E*). To exclude an artifact related to Lipofectamine transduction into cells, we tested FL (2A) tau preparations in an in vitro seeding assay that induces fibril formation by full-length tau (0N4R) through iterative polymerization and agitation steps (*Morozova et al., 2013*). $M_i$ had no intrinsic seeding activity. However $M_s$ induced amyloid formation, albeit more slowly than trimer or unfractionated fibrils (*Figure 1F*). This slow aggregation process may reflect inefficient fibril assembly, and a predominance of small nucleated assembly events from the added monomer. We concluded that the $M_s$ fraction contained seeding activity that

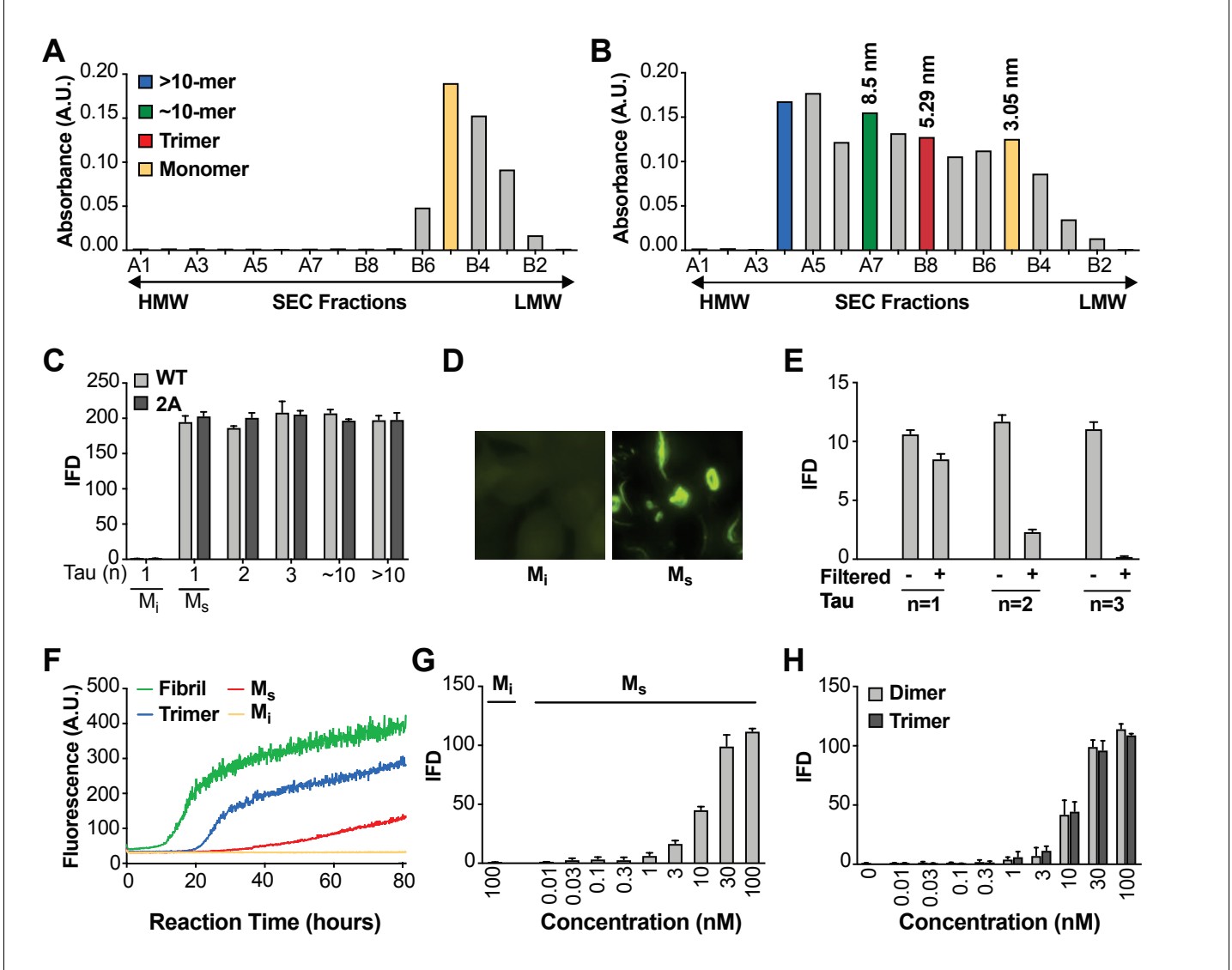

**Figure 1.** Seeding activity of tau monomer in cells and in vitro. (A, B) FL Cys-Tau(2A) was labeled with Alexa488 and resolved by SEC (A), or was fibrillized in the presence of heparin, labeled with Alexa488, sonicated, and the assemblies resolved by SEC (B). The column was calibrated using standards of the indicated hydrodynamic radii. Color codes indicate the putative assembly units. (C) Tau assemblies were seeded into tau RD-CFP/YFP biosensor cells. $M_i$ represents 'inert' monomer purified in (A), which had no seeding activity; $M_s$ represents 'seed-competent' monomer purified in (B), which induced intracellular tau aggregation (p<0.001). (D) FL WT tau and FL Cys-Tau(2A) were similarly fibrillized, sonicated, and the fragments resolved by SEC. Seeding activity of each fraction was determined. $M_s$ and larger assemblies of both forms of tau exhibited seeding activity, but not $M_i$. IFD = Integrated FRET Density. (E) Tau assemblies of n = 1,2,3 were passed through a 100kD size cutoff filter. Filtration had no effect on the $M_s$ fraction, whereas it reduced seeding of assemblies of n = 2 or 3. (F) Tau fibrils, trimer, or monomer were used to induce fibrillization in vitro of full-length (0N4R) tau, measured by induced thioflavin fluorescence. $M_i$ had no seeding activity, whereas $M_s$, trimer, and unfractionated fibrils had strong seeding activity. (G,H) Titration of assemblies was performed. (G) $M_s$ exhibited an $EC_{50}$ of approximately 10 nM (monomer equivalent); (H) Dimer and trimer had similar potencies. Concentration is reflected as monomer equivalent. See *Figure 1—source data 1*.

DOI: https://doi.org/10.7554/eLife.36584.003

The following source data is available for figure 1:

**Source data 1.** Data for $M_i$ and $M_s$ seeding activity in cells and in vitro.
DOI: https://doi.org/10.7554/eLife.36584.004

enabled intracellular aggregation of tau RD-CFP/YFP in cells, or full-length tau in vitro. Finally, we tested whether contamination of very small amounts of seeds could somehow account for the seeding activity in monomer fractions by carrying out dose-response titrations of the various preparations. $M_s$ had an $EC_{50}$ of ~10 nM (*Figure 1G*), which was very similar to dimer and trimer (*Figure 1H*). Thus to account for signal observed in the seeding assay, contamination of an otherwise inert monomer with larger seed-competent assemblies would have to be substantial.

## Comparison of $M_i$ and $M_s$ by CD and FCS

We tested for obvious structural differences between $M_i$ and $M_s$ using CD spectroscopy, which revealed none (*Figure 2A*). We re-tested the assemblies using fluorescence correlation spectroscopy (FCS), which measures particle diffusion through a fixed volume. As we previously observed (*Mirbaha et al., 2015*), we accurately estimated the units of small assemblies ($\leq$10 mer), but not larger assemblies (>10 mer) (*Figure 2B*). In an additional effort to detect cryptic multimers within the $M_s$ preparation, we used double-label FCS. We engineered a cysteine onto the amino terminus of FL tau (2A) to enable its covalent modification (Cys-Tau (2A)). We then prepared Cys-tau (2A) fibrils, or monomer, and labeled them simultaneously with Alexa488 (green) and tetramethylrhodamine (TMR) via maleimide chemistry. We carried out sonication and purification by SEC as before, isolating assemblies of various sizes. We evaluated each for cross-correlation between red and green signal, which indicates the presence of at least two tau molecules in a particle. We analyzed >300 events for each assembly. When we evaluated $M_i$ and $M_s$, 100% of events in each case showed a diffusion time consistent with a tau monomer (*Figure 2C,D*). Furthermore, we observed no cross-correlation between red and green signal, indicating that neither preparation had detectable multimeric assemblies (*Figure 2C,D,H*). By contrast, when we evaluated larger species such as dimer, trimer, or ~10 mer, we observed longer diffusion times consistent with the predicted assembly sizes, and significant cross-correlation values (*Figure 2E–H*), consistent with the presence of multimers. The FCS studies supported the conclusion that $M_i$ and $M_s$ are comprised predominantly of monomer.

## SEC preparation efficiently purifies $M_s$ monomer

To rule out cross-contamination of assemblies within the SEC column, we tested its ability to exclude larger seeds from the monomer fraction. We first isolated $M_s$ and larger assemblies from a sonicated fibril preparation (*Figure 3*, Group 1). Removing the fraction that contained $M_s$ (B5), we then pooled the remaining fractions, and spiked them with $M_i$. We re-fractionated the material on SEC to isolate the monomer in fraction B5 again (*Figure 3*, Group 2). As previously observed, $M_s$ and other fibril-derived assemblies in Group 1 had seeding activity (*Figure 3*). However, in Group 2, while we observed seeding activity in larger assemblies, the monomer (which we take to be $M_i$) re-isolated from a pool of larger fibril-derived assemblies had no seeding activity (*Figure 3*). This confirmed that larger, seed-competent assemblies do not appreciably contaminate the monomer fraction during SEC.

## Heat denaturation of assemblies

Although prior controls had essentially excluded the presence of tau multimers in the sample, we used heat-mediated dissociation of oligomeric assemblies as an additional test for the possibility that $M_s$ in fact represents a uniquely compact multimer that somehow purifies as a monomer. We collected $M_s$ by SEC, and heated the sample to 95°C for 3 hr. We then re-isolated the sample via SEC. We carried out the same procedure with trimer and ~20 mer. In each case, we tested the resultant fractions for seeding activity. In the first instance, after heating we re-isolated $M_s$ purely as monomer that retained virtually all of its seeding activity (*Figure 4A*). The trimer assembly (fraction B8) broke down to smaller assemblies, predominantly monomer, each of which retained seeding activity (*Figure 4B*). The ~20 mer (fraction A5) was largely stable following heat treatment, and retained its seeding activity (*Figure 4C*). These experiments highlighted the lability of small multimers (i.e. trimer), and a surprising persistence of seeding activity in heat-treated monomer.

## Differential heat lability of tau assemblies

In the preceding experiment $M_s$ retained seeding activity even after 3 hr at 95°C, a condition sufficient to dissociate trimers. These experiments implied that $M_s$ consists of a stable seed-competent

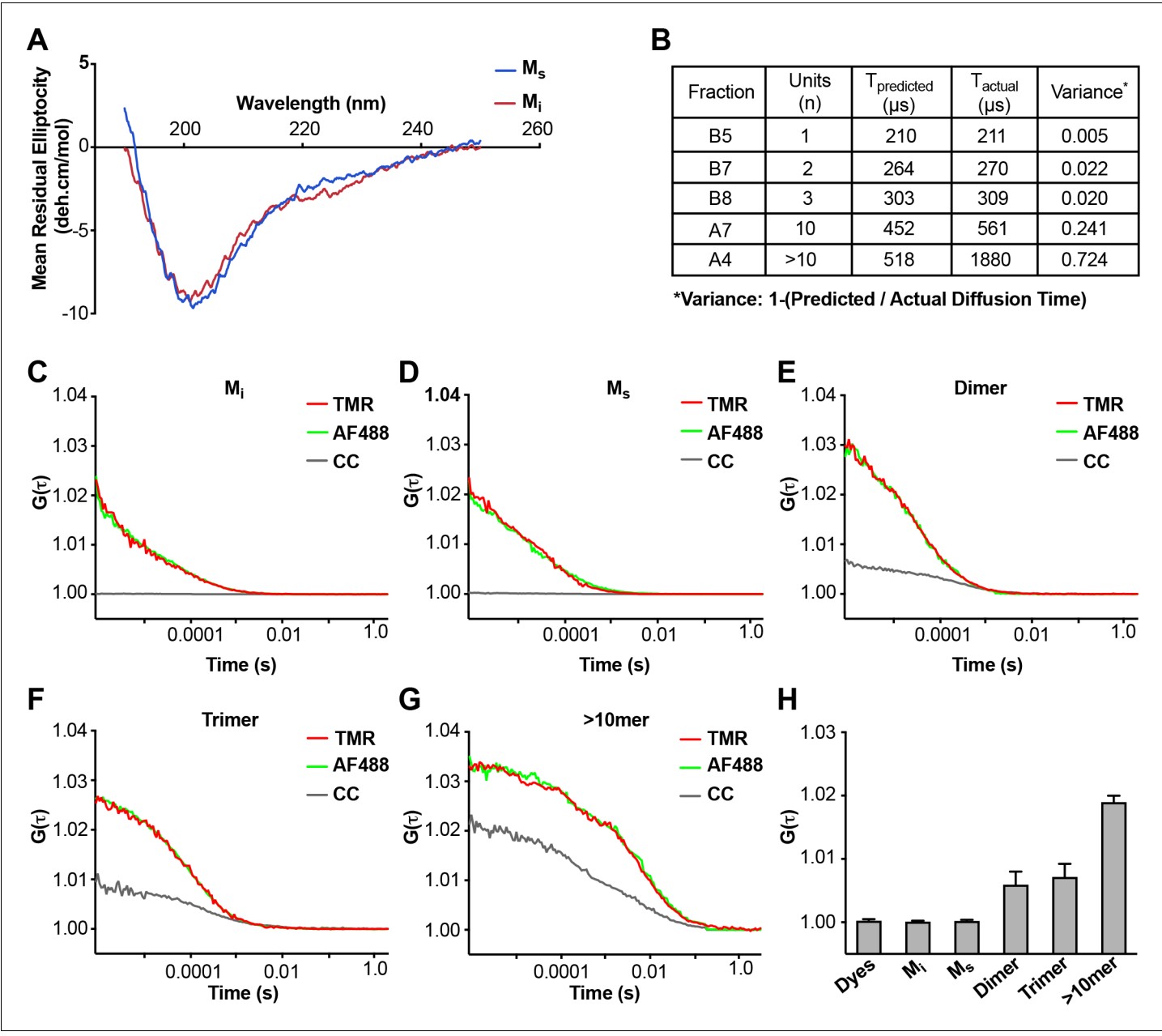

**Figure 2.** Analyses of $M_i$ and $M_s$ by CD and FCS. (**A**) CD spectra of $M_i$ and $M_s$ were similar. (**B**) FCS Diffusion times for $M_i$, $M_S$, dimer, trimer, and ~10 mer, and the cross-correlation for $M_i$, $M_s$, dimer, trimer, and $\geq$10 mer were determined after labeling of fibrils with Alexa488, or double labeling additionally with tetramethylrhodamine prior to sonication. Table reflects the predicted diffusion time and the actual diffusion time. The variance between predicted vs. observed times is reported. (**C–G**) FCS for double-labeled tau assemblies. Cross correlation (CC) between the two dyes is indicated in grey lines. (**H**) Summary of FCS cross-correlation, including free dyes. Neither free dye, $M_i$ nor $M_s$ showed any cross-correlation, indicating that single species predominate. All multimeric assemblies exhibited cross-correlation, indicating detection of both dyes within a single particle. See *Figure 2—source data 1*.

DOI: https://doi.org/10.7554/eLife.36584.005

The following source data is available for figure 2:

**Source data 1.** Data for CD and FCS of $M_i$ and $M_s$.

DOI: https://doi.org/10.7554/eLife.36584.006

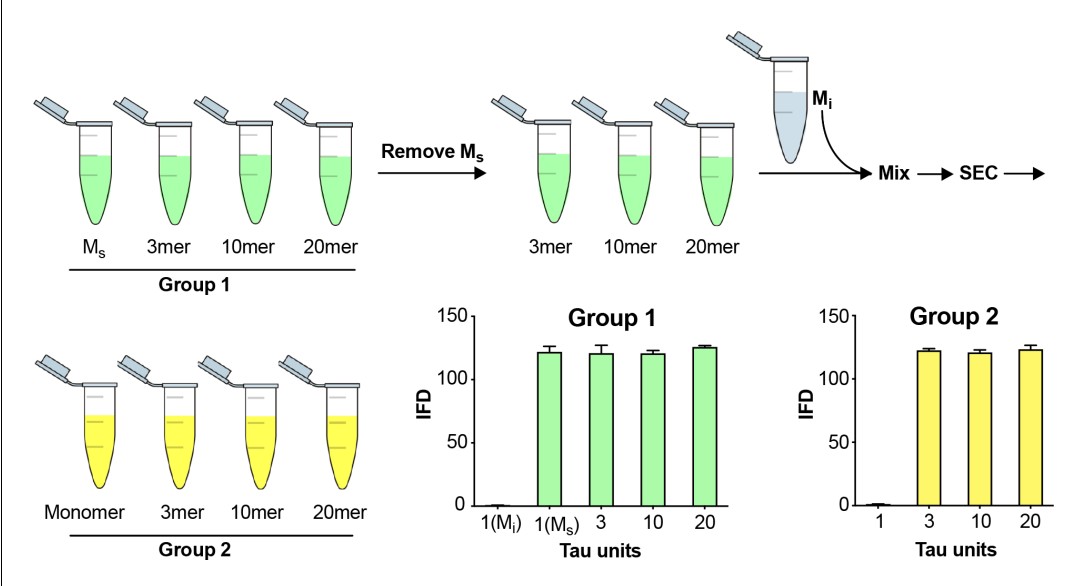

**Figure 3.** Fidelity of SEC purification of assemblies. SEC fidelity was tested by isolating $M_s$ from fractions after fibril sonication. Remaining fractions were combined with $M_i$, and the mix was re-isolated by SEC. In Group 1, after the first isolation, the monomer fraction (which contains $M_s$) contained seeding activity. In Group 2, after the second purification by SEC, the monomer fraction (which contains $M_i$ spiked in) did not exhibit seeding activity. See *Figure 3—source data 1*.

DOI: https://doi.org/10.7554/eLife.36584.007
The following source data is available for figure 3:

**Source data 1.** Data for fidelity of SEC purification of assemblies.
DOI: https://doi.org/10.7554/eLife.36584.008

structure, resistant to heat denaturation. Consequently, we used more nuanced heat denaturation of seeding activity to probe the relative stabilities of $M_s$, dimer, trimer, and larger assemblies of FL WT tau. We first isolated tau monomer, dimer, trimer,~10 mer, and ~20 mer on SEC. We then incubated the various assemblies at a range of temperatures (65, 75, 85, 95°C) and times (0, 3, 12, 18, 24, 48, 72 hr) before measuring seeding activity. Lower temperatures only slightly reduced seeding activity, whereas exposure of $M_s$, dimer, and trimer to temperatures $\geq$ 85°C for 18–24 hr eliminated it at roughly the same rate for each (*Figure 4D–G*). By contrast, the seeding activities of ~10 mer and ~20 mer were relatively heat-resistant (*Figure 4D–G*). This was consistent with our prior observations that tau seeds derived from cultured cells are resistant to boiling (*Sanders et al., 2014*). To determine a putative energy barrier between $M_s$ and $M_i$, we evaluated the denaturation data for $M_s$ by integrating the data from the prior experiments (*Figure 4H*). We compared two models for the transition of $M_s$ to an inert form (which we assumed to be an unfolding reaction): a unimodal unfolding model vs. a multimodal model that assumes intermediate seed-competent states. The unimodal model did not account for the data at early time points, which indicated a lag phase in denaturation, whereas the multimodel model performed better (*Figure 4H*). The lag phase in denaturation implied an ensemble of seed-competent states that define $M_s$, each separated by smaller energy barriers. Using the multimodal model, we calculated the barrier to conversion of $M_s$ to an inert form to be ~78 kcal/mol.

## $M_s$ has unique properties of self-assembly

Aggregation of $M_i$ in vitro is relatively slow, requires high protein concentration (micromolar), and polyanions such as heparin (*Goedert et al., 1996*; *Pérez et al., 1996*). Based on the seeding activity of $M_s$, we predicted that it might more readily self-associate. We incubated FL WT tau $M_i$ and $M_s$ alone, or dimer or trimer at equimolar ratios, keeping total particle concentration constant at 500 nM. We then monitored change in assembly size over 24 hr. $M_i$, dimer, and trimer showed no evidence of self-association in this timeframe (*Figure 5A,C,D*). By contrast, when incubated alone, $M_s$

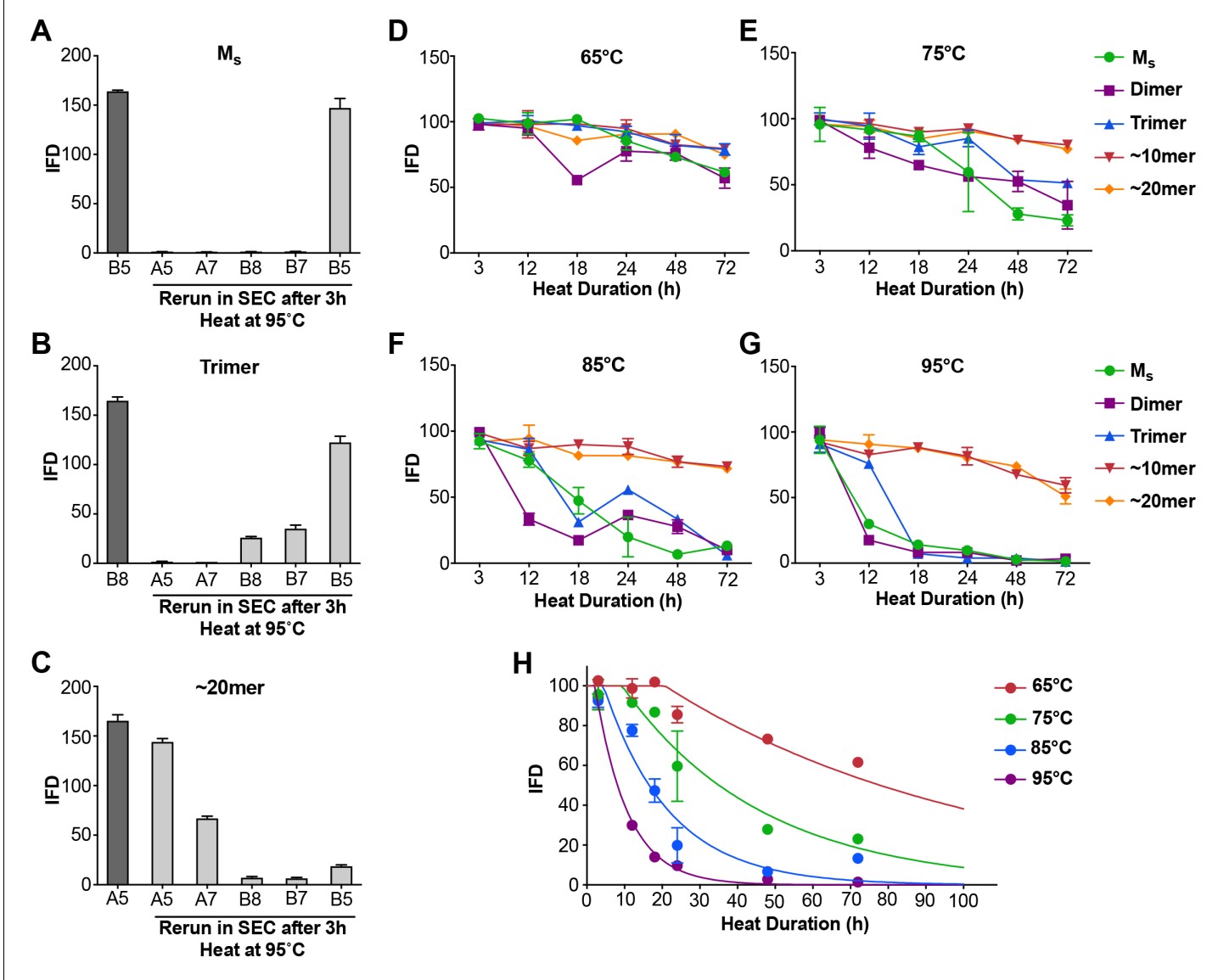

**Figure 4.** Heat denaturation of assemblies. (**A–C**) Heat-induced dissociation of assemblies. (**A**) The SEC fraction containing $M_s$ (B5) was heated to 95°C for 3 hr and re-isolated by SEC prior to testing the FRET biosensor assay. No loss in seeding activity was observed. (**B**) When the SEC fraction containing trimer (B8) was heated similarly, seeding activity shifted to fractions that contain dimer and monomer (B7, B5). (**C**) ~ 20 mer (A5) was largely stable to heating, although some smaller seed-competent assemblies were liberated. (**D–G**) Various assemblies were subjected to heat denaturation at the indicated temperatures and times, followed by analysis of seeding activity in the FRET biosensor assay. Whereas ~10 mer and ~20 mer were relatively stable from 65–95°C, monomer, dimer and trimer showed temperature-dependent loss of seeding activity. (**H**) Plot of denaturation data for $M_s$ with multimodal regression curves superimposed. See **Figure 4—source data 1**.

DOI: https://doi.org/10.7554/eLife.36584.009

The following source data is available for figure 4:

**Source data 1.** Data for heat denaturation of assemblies.

DOI: https://doi.org/10.7554/eLife.36584.010

readily formed larger assemblies (**Figure 5B**). When we incubated $M_i$ with dimer or trimer, we saw no change in the assembly population over 24 hr (**Figure 5E,F**). By contrast, when we mixed $M_s$ with dimer or trimer we observed a growth of larger assemblies with a concomitant reduction in dimer and trimer peaks (**Figure 5G,H**). We conclude that $M_i$, dimer, and trimer do not form larger assemblies at an appreciable rate, while $M_s$ self-assembles and adds on to larger assemblies.

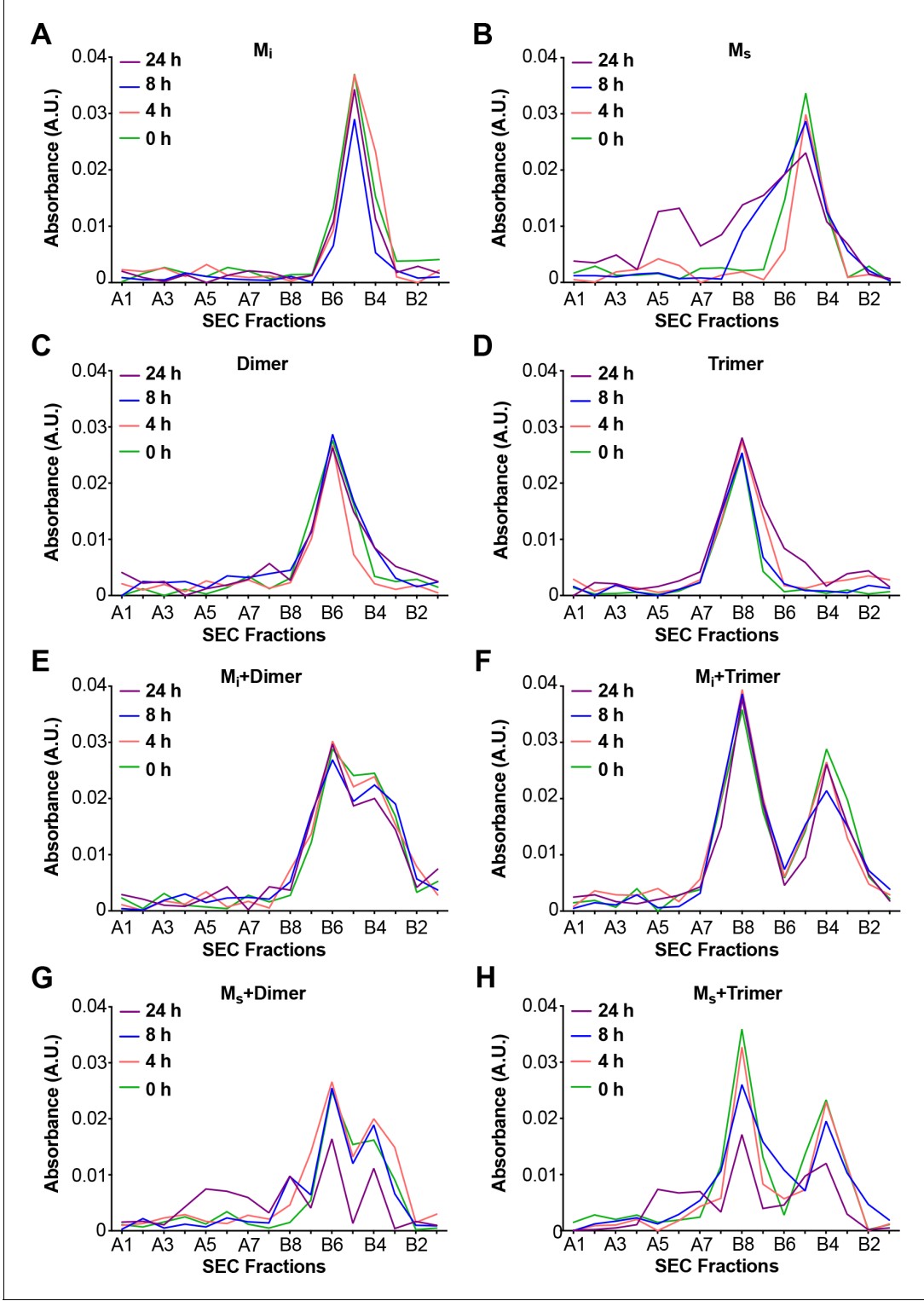

**Figure 5.** $M_s$ self-assembles. $M_i$ and $M_s$ were incubated at 500 nM or with equivalent amounts (monomer equivalent) of dimer and trimer for various times prior to resolution by SEC. Assemblies were monitored by reading the absorbance of fractions using micro BCA assay. (**A**) $M_i$ showed no self-association. (**B**) $M_s$ exhibited self-association over time. (**C,D**) Dimer and trimer were stable over time. (**E,F**) $M_i$ does not react with dimer or trimer to form larger assemblies. (**G,H**) $M_s$ reacts with dimer and trimer to form larger assemblies. See *Figure 5—source data 1*.

DOI: https://doi.org/10.7554/eLife.36584.011

The following source data is available for figure 5:

*Figure 5 continued on next page*

*Figure 5 continued*

**Source data 1.** Data for $M_s$ self-assembly.

DOI: https://doi.org/10.7554/eLife.36584.012

## Heparin induces transition from $M_i$ to $M_s$

The preparation of $M_s$ based on sonication of fibrils raised two important issues. First, it left uncertain whether $M_i$ could be converted to a seed-competent form without previously being incorporated into a fibril. Second, we observed that sonication could create fragments from tau monomer that might potentially act as seeds (*Figure 6—figure supplement 1*). Consequently, we used heparin to induce the formation of $M_s$, thereby avoiding sonication. We exposed FL WT tau to heparin for varying amounts of time before purifying different assembly sizes by SEC and testing for seeding activity. After 15 min of heparin exposure, we detected low but significant amounts of seed-competent monomer, and much fewer larger assemblies (*Figure 6A*). Crosslinking of purified, heparin-

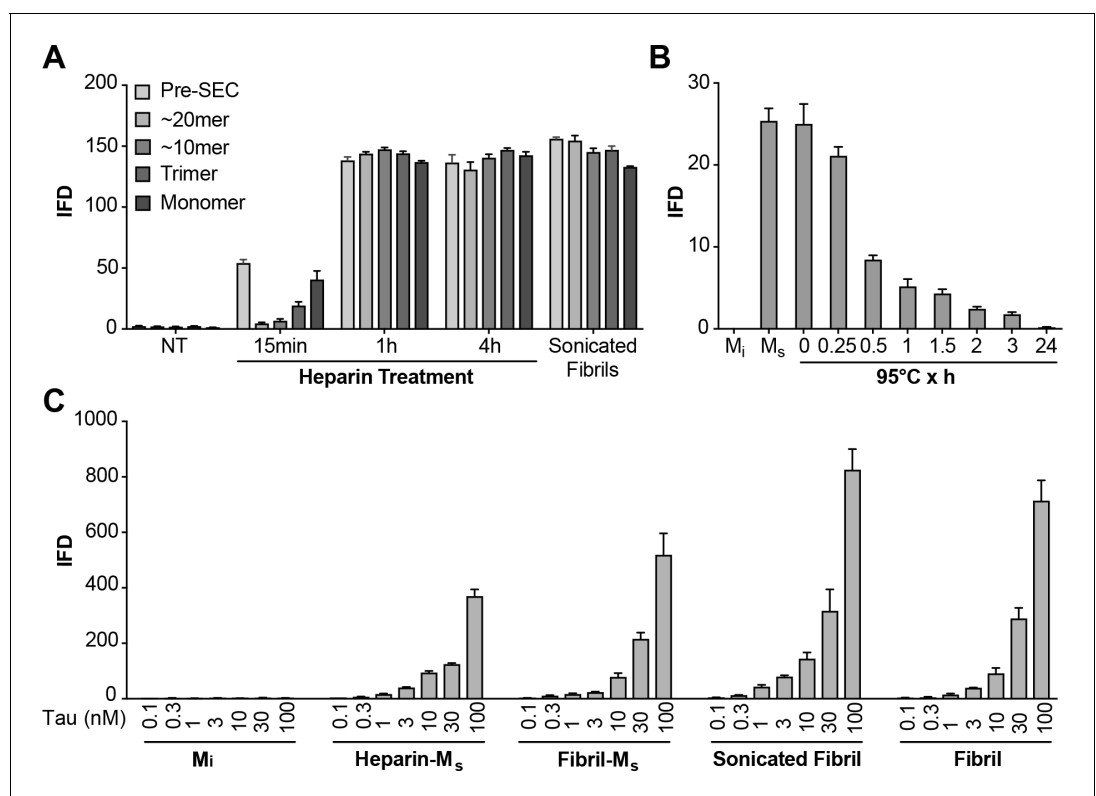

**Figure 6.** Heparin induces transition from $M_i$ to $M_s$. (**A**) Heparin treatment of FL WT tau was carried out for 15 min, 1 hr, or 4 hr. Samples were resolved by SEC, and fractions of various sizes were compared using the biosensor seeding assay. 'Pre-SEC' refers to the sample prior to fractionation. NT = monomer not treated with heparin. At 15 min, a small, but significant seeding activity was observed primarily in the monomer fraction. By 1 hr this signal was very strong, and comparable to the signal of $M_s$ derived from sonicated fibrils. (**B**) $M_s$ derived from 4 hr heparin exposure was heated at 95°C for different times, followed by analysis of seeding activity in the FRET biosensor assay. Seeding activity decayed over 24 hr. (**C**) Seeding efficiencies per nM of tau (monomer equivalent) of the various forms of $M_s$, sonicated, or unsonicated fibrils were relatively similar. $M_i$ was sonicated identically to $M_s$, followed by purification via SEC, but exhibited no seeding activity. Transfection of heparin failed to trigger intracellular aggregation (data not shown). See *Figure 6—source data 1*.

DOI: https://doi.org/10.7554/eLife.36584.013

The following source data and figure supplement are available for figure 6:

**Source data 1.** Data for heparin induced transition from $M_i$ to $M_s$.

DOI: https://doi.org/10.7554/eLife.36584.015

**Figure supplement 1.** SDS-PAGE of tau after sonication or heparin treatment.

DOI: https://doi.org/10.7554/eLife.36584.014

induced $M_s$ revealed no evidence of multimers or an increase in fragments (*Figure 6—figure supplement 1*). Recombinant monomer not treated with heparin had no seeding activity at any time point (*Figure 6A*). At longer heparin treatment times (1 hr, 4 hr) monomer fractions as well as larger assemblies all had strong seeding activity (*Figure 6A*). $M_s$ derived from heparin exposure was relatively resistant to heat denaturation at 95°C, albeit less so than fibril-derived $M_s$ (*Figure 6B*). Relative seeding efficiency of the various forms of $M_s$ as well as sonicated or unsonicated fibrils were relatively similar (*Figure 6C*). We noted also that sonication of $M_i$ and purification by SEC did not produce any seed-competent species, eliminating the possibility that small assemblies of sonication-induced fragments accounted for seeding activity of $M_s$ (*Figure 6C*). These experiments also indicated that it is not necessary for tau monomer to be part of a fibril or to be exposed to sonication to produce an efficient seed-competent monomer. Heparin, presumably by catalyzing a transition from an inert to a seed-competent form, enables this critical conformational change.

## XL-MS reveals unique contacts associated with $M_s$

To probe the structures of $M_i$ and $M_s$, we employed cross-linking with mass spectrometry (XL-MS), which uses DSS-mediated crosslinking of proteins (monomer or larger assembly) followed by trypsin proteolysis, enrichment of resultant fragments by SEC, and identification by capillary liquid chromatography tandem mass spectrometry. This method creates restraints for structural models of single proteins or protein complexes (*Leitner et al., 2012*; *Lasker et al., 2012*; *Joachimiak et al., 2014*). We assigned the complex fragment ion spectra to the corresponding peptide sequences using xQuest (*Rinner et al., 2008*). Denaturation of recombinant tau with 8M urea prior to crosslinking produced no intramolecular cross-links (data not shown), indicating that crosslinks observed under native conditions represented local structure. We studied $M_i$, fibril-derived $M_s$ and heparin-derived $M_s$ using XL-MS. Short reaction times ensured the production of only intra-molecular crosslinks as monitored by SDS-PAGE (*Figure 6—figure supplement 1*). XL-MS for each sample was carried out in triplicate , and only considering consensus crosslinks present in each replicate . $M_i$ exhibited crosslink patterns which indicated local and distant intramolecular contacts (*Figure 7A*). In $M_s$, we observed a consistent crosslinking of K150 with K254, K267, K274 or K280 all located between RD 1 and 2. These crosslinks tracked exclusively with $M_s$, both fibril- or heparin-derived (*Figure 7B,C*). We never observed these crosslinks in $M_i$. To test the relationship of this crosslink with seed function, we carried out heat denaturation at 95°C for 3 or 24 hr, followed by XL-MS. Heating samples results in a decrease in crosslink frequency (*Figure 7—figure supplement 1*). Importantly, however, we observed a parallel persistence of this crosslink pattern with seeding activity (*Figure 7B,C*). The XL-MS results indicate a distinct structure and seeding activity for $M_s$ that is surprisingly resistant to denaturation at 95°C.

## AD brain contains $M_s$

Given our experiments with recombinant $M_i$ and $M_s$, we wished to test whether similar structures exist in vivo. We extracted AD and control brain samples using a dounce homogenizer to avoid liberating significant monomer from fibrils. We immunoprecipitated tau using an antibody that targets the amino-terminus (HJ8.5), and resolved the eluates by SEC, followed by ELISA to determine tau levels (*Figure 8A,B*). Tau from control brain purified in the monomer fraction (*Figure 8A*), while tau from AD brain distributed across multiple fractions, corresponding to monomer and larger assemblies (*Figure 8B*). When we tested each fraction for seeding activity, we observed none in any control brain fraction (*Figure 8C*). However, all AD fractions contained seeding activity, including monomer (*Figure 8C*). To exclude the possibility that the brain homogenization protocol liberated $M_s$ from neurofibrillary tangles, we spiked tau KO mouse brain samples with recombinant fibrils in vitro, or fibril-derived $M_s$. We then used dounce homogenization and immuno-purification as for human brain. We evaluated the seeding activity in total lysate, supernatant following 10,000 x g centrifugation, and SEC fractions (*Figure 8D*). We readily observed monomer seeding activity in tau KO brain spiked with $M_s$, however we observed none in fractions that had been spiked with fibrils (*Figure 8D*). The homogenization protocol for human brain was thus unlikely to have liberated $M_s$ from pre-existing tau fibrils.

To test for self-association of control-derived $M_i$ vs. AD-derived $M_s$, we purified these species by SEC, and divided each monomer fraction in two. We snap-froze one fraction and incubated the

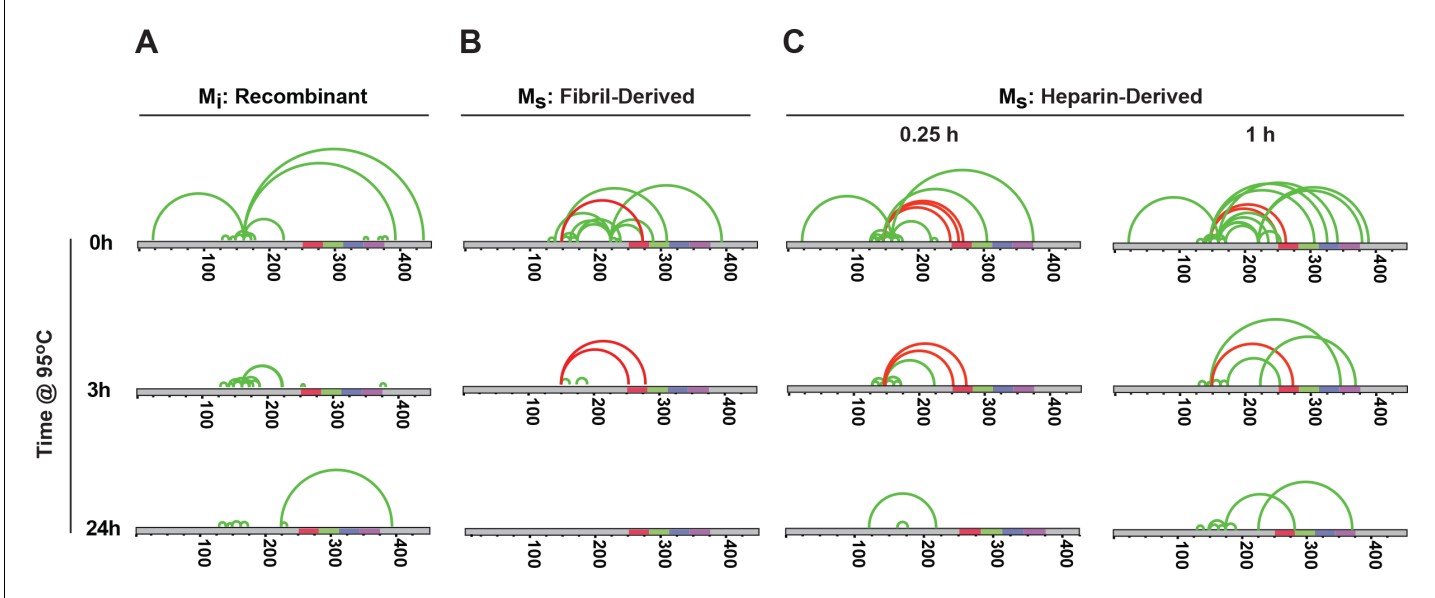

**Figure 7.** Unique XL-MS patterns for different forms of $M_i$ and $M_s$. Tau monomers were prepared as described, heated at 95°C for 0, 3 or 24 hr, reacted with DSS, proteolyzed and analyzed by mass spectrometry to define intramolecular crosslinks. Diagrams represent crosslinks within the tau protein. Tau is shown in grey; RD is colored in red (R1), green (R2), blue (R3) and indigo (R4). Each diagram indicates consensus crosslinks present across replicates (N = 3) (green or red). Crosslinks uniquely observed within $M_s$ preparations are shown in red. Each sample was prepared, isolated by SEC, and then subjected XL-MS. (A) $M_i$: tau monomer not previously fibrillized; (B) $M_s$: fibril-derived tau monomer; (C) $M_s$: heparin-exposed tau monomer (0.25 hr or 1 hr). Crosslinks from aa150 to aa254-290 mark all forms of $M_s$ after exposure to 95°C for 0 hr, 0.25 hr and 3 hr, but not 24 hr. See **Figure 7—source data 1** and **2**.

DOI: https://doi.org/10.7554/eLife.36584.016

The following source data and figure supplement are available for figure 7:

**Source data 1.** Summary of triplicate XL-MS datasets.
DOI: https://doi.org/10.7554/eLife.36584.018
**Source data 2.** Summary of consensus XL-MS datasets.
DOI: https://doi.org/10.7554/eLife.36584.019
**Figure supplement 1.** Frequency of crosslinks decreases with heat incubation.
DOI: https://doi.org/10.7554/eLife.36584.017

other overnight at room temperature. Then, we again resolved the assemblies via SEC and tested each fraction for seeding activity. Control monomer was inert, even after incubation at RT (**Figure 8E**). AD-derived $M_s$ that was purified, frozen, and re-purified by SEC exhibited seeding activity exclusively in the monomer fraction (**Figure 8E**). By contrast, AD-derived $M_s$ incubated at RT formed seed-competent assemblies of increasing size (**Figure 8E**). We concluded that, as for other types of $M_s$, AD-derived $M_s$ exhibited an intrinsic capacity for self-association into seed-competent assemblies. To compare structures of control vs. AD-derived monomer via XL-MS, we isolated tau from brains of 3 AD patients and three age-matched controls. In control-derived monomer, we observed no evidence of the crosslink that marked $M_s$ (**Figure 8G**). However, in each AD-derived $M_s$ sample we observed a discrete set of crosslinks between aa150 and aa259-290 (**Figure 8H**). This essential finding did not change, no matter what method of homogenization we used (**Figure 8—figure supplement 1**), and implied a common structure that unifies ensembles of seed-competent tau monomer, whether produced in vitro or in vivo.

## Models of seed-competent monomer suggest exposure of VQIINK and VQIVYK

Based on intramolecular FRET and electron paramagnetic resonance spin labeling Mandelkow et al. have previously proposed native tau structure to be in a 'paperclip' configuration, with the C-terminus folded over the RD (**Jeganathan et al., 2006**). To understand how core elements of tau control its aggregation, we employed Rosetta to create models of tau structure for $M_i$ and $M_s$ using

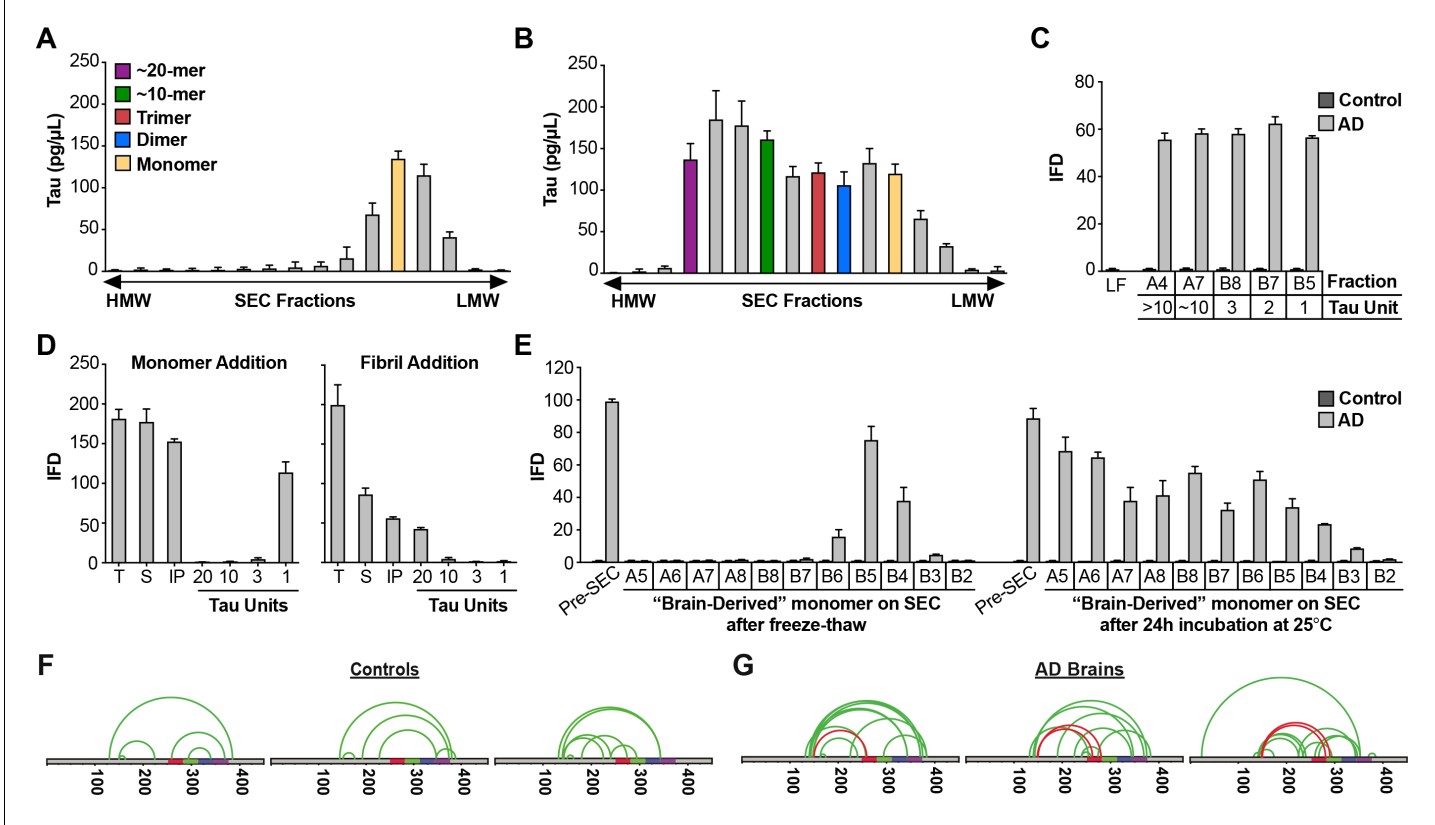

**Figure 8.** AD brain contains seed-competent monomer. Tau from control and AD brains was immunoprecipitated and subjected to SEC. (A) SEC from control brain contained predominantly tau monomer. (B) SEC from AD brain contained a range of tau assembly sizes. (C) Tau monomer from control brain exhibited no seeding activity, whereas monomer from AD brain did, along with larger assemblies (p<0.001). Tau Unit refers to the putative number of molecules per assembly. LF = Lipofectamine control. (D) Tau KO mouse brain was spiked either with human tau $M_s$ or fibrils prior to dounce homogenization, immunopurification, and resolution by SEC. Samples spiked with $M_s$ exhibited monomer seeding activity, but not samples that had been spiked with fibrils. (E) AD-derived tau monomer was incubated for the indicated times prior to SEC and determination of seeding activity in each fraction. Larger seed-competent assemblies formed after 24 hr incubation at RT. (F, G) Three control and AD brains were homogenized, monomer isolated, and evaluated by XL-MS. Tau monomer from controls lacked the long-range crosslinks observed in $M_s$. AD-derived $M_s$ contained long-range crosslinks (aa150 to aa254-290) also observed in recombinant forms of $M_s$. See *Figure 8—source datas 1* and *2*.

DOI: https://doi.org/10.7554/eLife.36584.020

The following source data and figure supplement are available for figure 8:

**Source data 1.** Data characterization of patient samples.
DOI: https://doi.org/10.7554/eLife.36584.022
**Source data 2.** Summary of patient-derived XL-MS datasets.
DOI: https://doi.org/10.7554/eLife.36584.023
**Figure supplement 1.** Different brain homogenization methods yield similar crosslink patterns.
DOI: https://doi.org/10.7554/eLife.36584.021

restraints from the crosslink patterns and length of the DSS crosslinker. The overall energetics and radii of gyration in the models were comparable for $M_i$ and $M_s$ (*Figure 9—figure supplement 1*), indicating global structural similarity. We thus focused on the RD, given its high frequency of intra-molecular crosslinks, and primary role in aggregation (*Figure 9A*). We observed differences in the predicted interface structure between R1/R2 and R2/R3 which encode two core VQIINK and VQIVYK motifs critical for tau amyloid formation (*von Bergen et al., 2000*; *von Bergen et al., 2001*). The $M_i$ structural model predicted masking of VQIINK and VQIVYK sequences in compact 'hairpin' structures (*Figure 9B*), similar to the structure of microtubule-bound tau previously determined by NMR (*Kadavath et al., 2015*). By contrast, within $M_s$ the model predicted relative exposure of VQIINK and VQIVYK (*Figure 9C*). We next evaluated XL-MS-guided predictions of patient-derived tau, although lower sample quality and fewer high confidence crosslinks (possibly due to protein

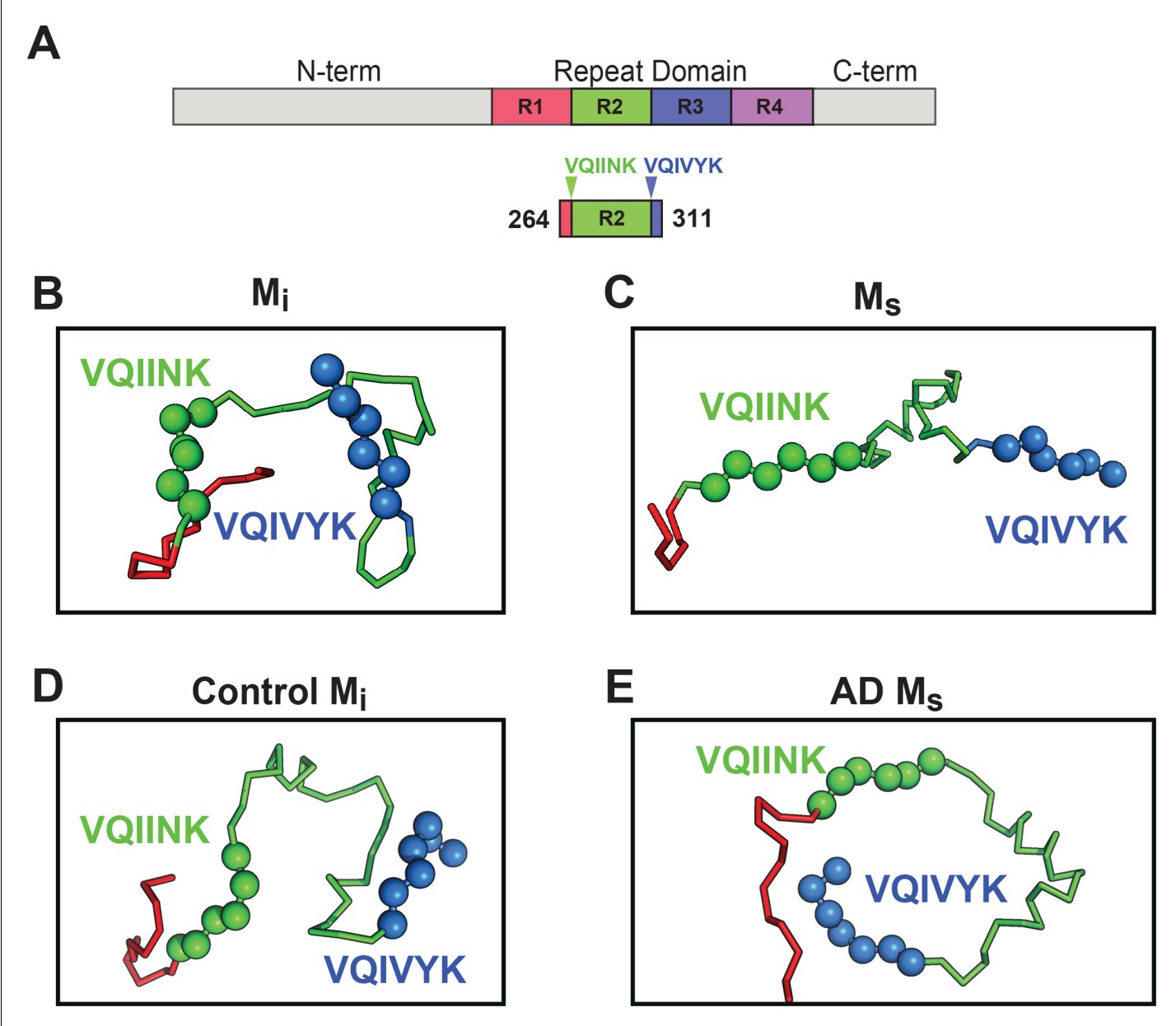

**Figure 9.** Models of $M_i$ and $M_s$ suggest differences in the R1R2 and R2R3 regions. XL-MS identified pairs were used as restraints in Rosetta to create structural models of discrete tau domains. (A) Schematic highlighting the region of the RD encoding structural differences between $M_i$ and $M_s$. Tau RD is colored in red (R1), green (R2), blue (R3) and indigo (R4); N- and C-terminal portions of tau are shown in grey. Fragments of interest are shown with their position in the RD. (B) recombinant $M_i$; (C) fibril-derived $M_s$, (D) Control $M_i$ and (E) AD-derived $M_s$. Regions surrounding the R1R2 and R2R3 are indicated, highlighting two amyloid-forming sequences, VQIINK (green spheres) and VQIVYK (blue spheres). In both forms of $M_i$ VQIINK and VQIVYK are associated with flanking amino acids in hairpin structures. In both forms of $M_s$ the VQIINK and VQIVYK sequences are presented at the protein surface. See *Figure 9—source data 1*.

DOI: https://doi.org/10.7554/eLife.36584.024

The following source data and figure supplement are available for figure 9:

**Source data 1.** Models of $M_i$, $M_s$, control and AD conformations of tau.
DOI: https://doi.org/10.7554/eLife.36584.026
**Source data 2.** Rosetta energy and radius of gyration for structural ensembles.
DOI: https://doi.org/10.7554/eLife.36584.027
**Figure supplement 1.** Energetics of Rosetta structural ensembles.
DOI: https://doi.org/10.7554/eLife.36584.025

heterogeneity) limited our accuracy. As for recombinant protein, $M_i$ from control patients also featured VQIINK/VQIVYK sequences in a less accessible configuration (*Figure 9D*). In AD-derived $M_s$, long-range contacts from aa150 to R2 influenced the model, and predicted an exposed configuration of VQIINK/VQIVYK (*Figure 9E*). With important caveats, the models guided by XL-MS imply that the general difference between $M_i$ and $M_s$ derives from relative shielding vs. exposure of VQIINK/VQIVYK sequences.

### Limited proteolysis supports models of exposed VQIINK/VQIVYK sequences

As an orthogonal comparison of the structures of $M_i$ and $M_s$, we used limited proteolysis with trypsin. $M_i$ or $M_s$ (heparin-exposed) that had been passed through a 100kD filter immediately prior were subjected to a fine time course of limited proteolysis (*Figure 10A*). Each sample was prepared in triplicate with matched protein quantities to facilitate label-free analysis. We then used mass spectrometry to evaluate the production of tau fragments and mapped these to specific cleavage sites (*Figure 10B*). We identified 60 peptides common across the two conditions (*Figure 10—figure supplement 1*). To summarize enrichment of peptides across the two datasets we compared the ratio of averaged kinetic profiles (*Figure 10—figure supplement 1*). Differences between the $M_i$ and $M_s$ primarily localized to the RD (*Figure 10—figure supplement 1*). In $M_i$, an R1R2 fragment was enriched (*Figure 10C*) while only the R2 portion of that fragment was enriched in $M_s$ (*Figure 10D*). We observed similar patterns in R2R3 (*Figure 10F,G*). By contrast, other domains outside of these regions had similar cleavage kinetics in $M_i$ and $M_s$ (*Figure 10E,H*, *Figure 10—figure supplement 1*). Mapping these cleavage sites onto our structural models revealed that proteolysis in $M_i$ preferentially occurred outside the hairpin that includes VQIINK and VQIVYK amyloid sequences, while cleavage in $M_s$ occurred adjacent to the amyloid sequences (*Figure 10I,J*). The cleavage patterns were thus consistent with structural models of VQIINK and VQIVYK regions, which predicted relative inaccessibility of hairpin-associated sequences in $M_i$, and accessibility in $M_s$.

## Discussion

We propose that tau monomer occupies two distinct and stable conformational ensembles. One set of structures (collectively termed $M_i$) is relatively inert, while another has intrinsic ability to self-assemble, and acts as a template, or seed, for fibril growth in vitro and in cells (collectively termed $M_s$). Multiple controls indicated that our original preparation of fibril-derived $M_s$ is in fact a monomer, uncontaminated by larger assemblies. Tau monomer purified from AD brain also had intrinsic seeding activity, and self-associated to produce larger seed-competent assemblies. A model restrained by the XL-MS data, and consistent with biochemical studies, predicts that VQIVYK and VQIINK sequences assume an open configuration in all types of $M_s$ (fibril-derived, heparin-induced, and AD-derived). By contrast, the model predicts lack of VQIINK/VQIVYK exposure in $M_i$. Limited proteolysis studies are consistent with this idea, although clearly more detailed biochemical, biophysical, and structural analyses will be needed to test its validity. Taken together, these data establish a new concept for tau: this intrinsically disordered protein has multiple, stable monomeric states, functionally distinguished by the presence or absence of seeding activity.

Amyloid proteins form progressively larger assemblies over time, and it has been difficult to define the composition of the minimal seed. Mandelkow and colleagues studied tau aggregation in vitro and concluded that a seed of 8–12 molecules existed in their experimental system (*Friedhoff et al., 1998*). By contrast, Kuret and colleagues posited an 'intermediate' of tau that could subsequently initiate self-assembly, and their data, based on extrapolation of tau concentrations needed to enable development of thioflavin fluorescence in vitro, were consistent with a monomeric seed (*Chirita et al., 2005*). Wetzel and colleagues also proposed that a monomer is the basis of a 'thermodynamic nucleus' that templates the aggregation of synthetic polyglutamine peptides (*Bhattacharyya et al., 2005*). However, no prior study has previously identified stable forms of tau monomer that seed amyloid formation.

The actual cause of tau aggregation in tauopathies is unknown. It has been proposed that dissociation of tau monomer from microtubules, possibly due to phosphorylation, allows high concentration and self-association to form pathogenic assemblies (*Mandelkow and Mandelkow, 2012*). In this study, using a single source of recombinant protein, we define distinctly structured seed-competent

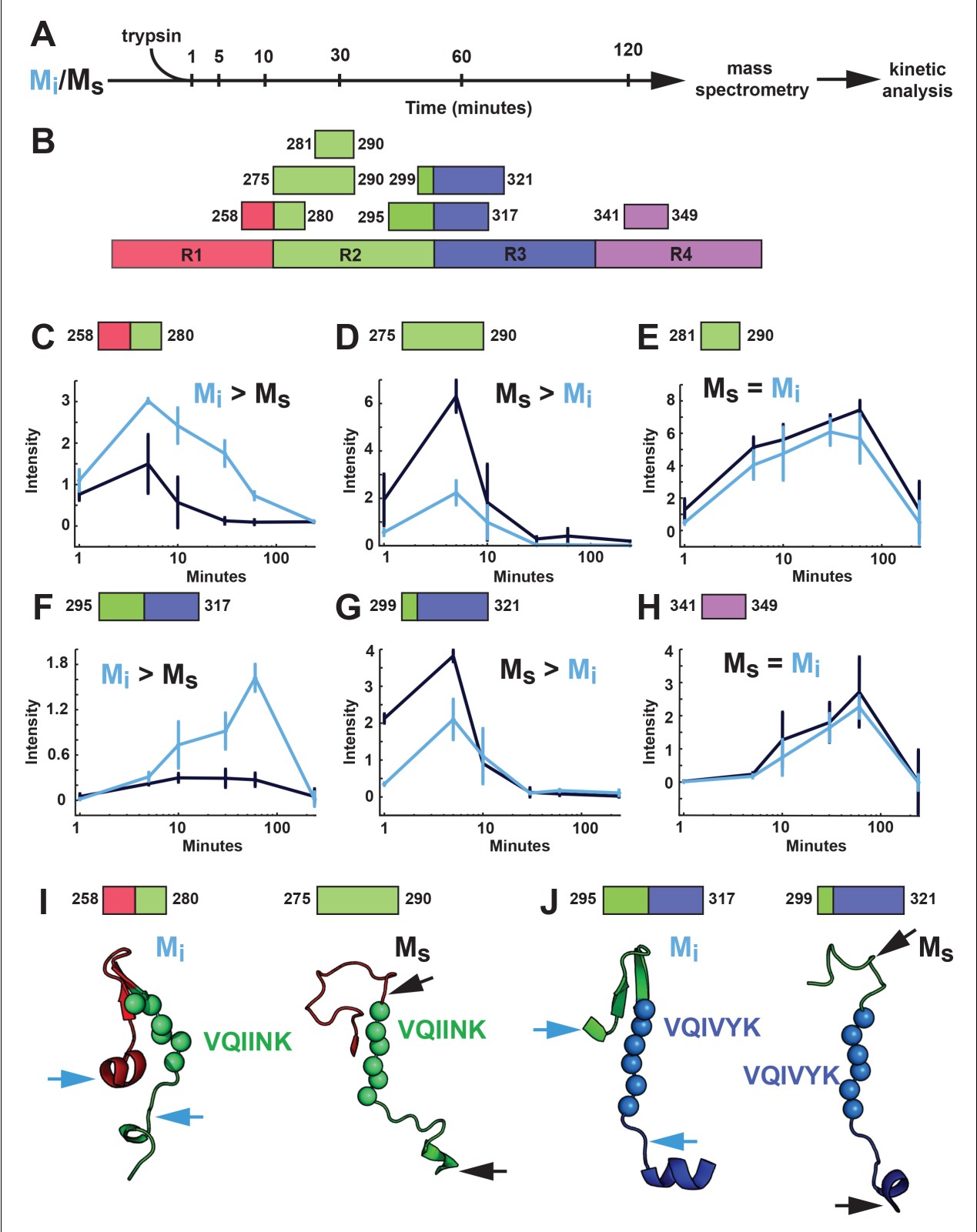

**Figure 10.** Proteolysis of $M_i$ and $M_s$ reveals distinct patterns. (A) $M_i$ and $M_s$ were prepared as technical triplicates (N = 3), isolated by SEC, and passed through a 100kD filter immediately prior to exposure to trypsin for 1, 5, 10, 30, 60 and 120 min. Samples were analyzed by mass spectrometry and kinetic profiles were generated for peptides present at each time point. (B) Tau RD is colored in red (R1), green (R2), blue (R3) and indigo (R4). Identified peptides are shown with their position in the RD. (C–H) Kinetic profiles are indicated for peptides that were more abundant in $M_i$ (C, F), $M_s$

*Figure 10 continued on next page*

*Figure 10 continued*

(D, G) or equal in $M_i$ and $M_s$ (E, H). $M_i$ and $M_s$ kinetic profiles are shown in blue and black, respectively. Fragments enriched in $M_i$ or $M_s$ were mapped onto corresponding regions in the structural models (I, J). The models are shown as cartoons colored in red (R1), green (R2) and blue (R3). Cleavage sites are indicated by arrows for $M_i$ (blue) and $M_s$ (black). See *Figure 10—source data 1*.

DOI: https://doi.org/10.7554/eLife.36584.028

The following source data and figure supplement are available for figure 10:

**Source data 1.** Summary of peptides and their abundance identified in the triplicate $M_i$ and $M_s$ proteolysis experiment.
DOI: https://doi.org/10.7554/eLife.36584.030
**Source data 2.** Summary of ratiometric analysis of $M_i$ and $M_s$ peptide abundance.
DOI: https://doi.org/10.7554/eLife.36584.031
**Figure supplement 1.** Proteolysis reveals localized differences between $M_i$ and $M_s$.
DOI: https://doi.org/10.7554/eLife.36584.029

and inert forms of tau. We have similarly identified seed-competent species in human brain. In reality 'seed-competent' and 'inert' forms of tau almost certainly represent multiple structural ensembles separated by defined energy and/or kinetic barriers. The barrier for conversion of an inert to a seed-competent form of tau can apparently be overcome by incubation with heparin and/or incorporation into a fibril. In neurons, other factors such as post-translational modifications and heterologous binding events likely play a role. Identification of the factors that trigger conversion from inert to seed-competent forms will thus have obvious implications for understanding disease mechanisms.

Isolation of seed-competent monomer from AD brain, with a very mild purification that explicitly excludes sonication or vigorous tissue homogenization, strongly suggests that this form of tau exists in vivo. Furthermore, we observed that both recombinant $M_s$ and AD-derived $M_s$ build multimeric assemblies in vitro far more efficiently than $M_i$ or control-derived monomer. Thus, we hypothesize that a uniquely structured form of tau may be required for efficient assembly growth in cells. This contrasts with the idea that multimeric assemblies uniquely stabilize the conformation of otherwise unstructured proteins as they incorporate into the growing fibril, or that liquid-liquid phase separation with extremely high local concentration underlies tau aggregation (*Wegmann et al., 2018*). Instead, we imagine that the initiation of aggregation in human brain might begin with a stable transition of tau monomer from an inert to a seed-competent form. To fully study this process will require more extensive biochemical purification of tau $M_s$ from the earliest stages of disease.

$M_s$ has a remarkably stable structure, as it resists heat denaturation at 95°C for up to 3 hr. This suggests a heretofore unrecognized conformation of tau that, to account for its slow denaturation, likely involves multiple intra-molecular interactions involving short and long range amino acid contacts. XL-MS provides some indication of what these might be, and crosslinks between aa150 and R1/R2 appear to mark a seed-competent conformation. In agreement with the XL-MS results, we observed that heat inactivation of $M_s$ seeding activity occurs with a lag phase, rather than first order time-dependent decay. This implies a complex tertiary structure in which $M_s$ has multiple seed-competent intermediates. Future XL-MS studies performed at different temperatures could reveal these structures. With more advanced methods to interrogate the structure of monomeric tau in patient material, we imagine that 'seed-competent monomer' will in fact represent myriad structures, depending on the underlying disease. This could provide an explanation for how a single tau protein might self-assemble into diverse amyloid strains. We note with excitement a recent study of the yeast prion Sup35 from the Tanaka laboratory. Like tau, Sup35 is intrinsically disordered, yet they have observed local structure that influences the conformations of fibrils it can form (*Ohhashi et al., 2018*).

Without further studies to identify structures of tau at higher resolution, we cannot know for certain why one form acts as a seed and another does not. However, we gained important insights when we modeled the configurations of R1R2 and R2R3 using Rosetta, with crosslinks as restraints. With obvious caveats, our models predicted that the local environment surrounding two hexapeptide motifs, VQIINK and VQIVYK, which are required for tau to form amyloid structures, may explain the differences between seed-competent and inert forms. In the models of $M_i$, and control brain-derived tau, these motifs lie buried in hairpin structures. By contrast, in $M_s$ and AD-derived tau, both are exposed. VQIINK and VQIVYK thus might mediate intermolecular interaction in a growing assembly. In support of our structural model, the proteolysis experiments corroborate differences in

exposure of the VQIINK and VQIVYK sequences in the R1R2 and R2R3 regions between $M_i$ and $M_s$. We note with great enthusiasm the recent study of Fitzpatrick et al. (*Fitzpatrick et al., 2017*), which defined critical sequences of tau within the amyloid core that are based on VQIVYK and adjacent amino acids. Indeed, it has been recently observed that heparin binding involves residues spanning 270–290, and promotes expansion of the remainder of the molecule (*Zhao et al., 2017*). This is consistent with our predictions of relative exposure of VQIINK/VQIVYK. The diversity of exposed core elements (almost certainly beyond VQIINK/VQIVYK) could specify the formation of assemblies that give rise to distinct strains, as suggested by work from the Tanaka laboratory (*Ohhashi et al., 2018*). Consistent with this idea, the Fitzpatrick et al. study indicates that in AD-derived tau fibrils the VQIVYK sequence plays a key role in the core amyloid structure (along with adjacent amino acids), but the VQIINK sequence does not (*Fitzpatrick et al., 2017*). We also note that multiple disease-associated mutations in tau affect residues in close proximity to VQIINK/VQIVYK. For example, our models predict that serine or leucine substitutions at P301 (which cause dominantly inherited tauopathy) would uniquely destabilize the local structure and promote exposure of the VQIINK/VQIVYK sequences. Future experiments will test these ideas more definitively.

## Materials and methods

**Key resources table**

| Reagent type (species) or resource | Designation | Source or reference | Identifiers | Additional information |
|---|---|---|---|---|
| Cell line (HEK293) | Tau RD P301S FRET Biosensor | Produced by Diamond lab, also available from ATCC | ATCC CRL-3275 | |
| biological sample (mouse) | Tau KO mouse | Jackson Laboratories | 7251 | |
| biological sample (human) | Alzheimer's Disease brain, stage VI, frozen | Washington University in St. Louis | 62579, 62732, 61473 | |
| biological sample (human) | Human normal brain | Washington University in St. Louis | 60974, 607, 64238 | |
| antibody | HJ8.5 | Produced by Diamond lab | N/A | Monoclonal antibody against human tau |
| antibody | Polyclonal Rabbit Anti-Human Tau | Dako, Denmark | A0024 | |
| antibody | ECL Anti-rabbit IgG Horseradish Peroxidase F(ab) fragment | GE Healthcare | NA9340V | |
| recombinant protein | Human tau 2N4R (Full Length WT-tau) | Produced by Diamond lab | N/A | MAEPRQEFEV MEDHAGTYGL GDRKDQGGYT MHQDQEGDTD AGLKESPLQT PTEDGSEEPG SETSDAKSTP TAEDVTAPLV DEGAPGKQAA AQPHTEIPEG TTAEEAGIGD TPSLEDEAAG HVTQARMVSK SKDGTGSDDK KAKGADGKTK IATPRGAAPP GQKGQANATR IPAKTPPAPK TPPSSGEPPK SGDRSGYSSP GSPGTPGSRS RTPSLPTPPT REPKKVAVVR TPPKSPSSAK SRLQTAPVPM |

*Continued*

| Reagent type (species) or resource | Designation | Source or reference | Identifiers | Additional information |
|---|---|---|---|---|
| | | | | PDLKNVKSKI GSTENLKHQP GGGKVQIINK KLDLSNVQSK CGSKDNIKHV PGGGSVQIVY KPVDLSKVTS KCGSLGNIHH KPGGGQVEVK SEKLDFKDRV QSKIGSLDNI THVPGGGNKK IETHKLTFRE NAKAKTDHGA EIVYKSPVVS GDTSPRHLSN VSSTGSIDMV DSPQLATLAD EVSASLAKQG L |
| recombinant protein | Tau (2A) | Produced by Diamond lab | N/A | MAEPRQEFEV MEDHAGTYGL GDRKDQGGYT MHQDQEGDTD AGLKESPLQT PTEDGSEEPG SETSDAKSTP TAEDVTAPLV DEGAPGKQAA AQPHTEIPEG TTAEEAGIGD TPSLEDEAAG HVTQARMVSK SKDGTGSDDK KAKGADGKTK IATPRGAAPP GQKGQANATR IPAKTPPAPK TPPSSGEPPK SGDRSGYSSP GSPGTPGSRS RTPSLPTPPT REPKKVAVVR TPPKSPSSAK SRLQTAPVPM PDLKNVKSKI GSTENLKHQP GGGKVQIINK KLDLSNVQSK AGSKDNIKHV PGGGSVQIVY KPVDLSKVTS KAGSLGNIHH KPGGGQVEVK SEKLDFKDRV QSKIGSLDNI THVPGGGNKK IETHKLTFRE NAKAKTDHGA EIVYKSPVVS GDTSPRHLSN VSSTGSIDMV DSPQLATLAD EVSASLAKQG L |
| recombinant protein | Cys-Tau (2A) | Produced by Diamond lab | N/A | MAEPRQEFEV MEDHACGTYGL GDRKDQGGYT MHQDQEGDTD AGLKESPLQT |

*Continued*

| Reagent type (species) or resource | Designation | Source or reference | Identifiers | Additional information |
|---|---|---|---|---|
| | | | | PTEDGSEEPG SETSDAKSTP TAEDVTAPLV DEGAPGKQAA AQPHTEIPEG TTAEEAGIGD TPSLEDEAAG HVTQARMVSK SKDGTGSDDK KAKGADGKTK IATPRGAAPP GQKGQANATR IPAKTPPAPK TPPSSGEPPK SGDRSGYSSP GSPGTPGSRS RTPSLPTPPT REPKKVAVVR TPPKSPSSAK SRLQTAPVPM PDLKNVKSKI GSTENLKHQP GGGKVQIINK KLDLSNVQSK AGSKDNIKHV PGGGSVQIVY KPVDLSKVTS KAGSLGNIHH KPGGGQVEVK SEKLDFKDRV QSKIGSLDNI THVPGGGNKK IETHKLTFRE NAKAKTDHGA EIVYKSPVVS GDTSPRHLSN VSSTGSIDMV DSPQLATLAD EVSASLAKQG L |
| commercial assay or kit | micro BCA Protein Assay Kit | Thermo Scientific | 23235 | |
| chemical compound | Heparin sodium salt from porcine intestinal mucosa | Sigma | H4784 | |

## Tau expression, purification, fibrillization, and labeling

We utilized several forms of recombinant tau. Full-length (FL), wild-type (WT) tau contains two cysteines that create disulfide bridges and could complicate isolation of monomer. Thus in addition to preparing FL WT tau (2N4R) as previously described (*Frost et al., 2009b*), we purified FL tau (2N4R) that contains two cysteine/alanine substitutions (C291A, C322A), termed tau (2A). We used the 2A and WT forms of tau in our initial studies, before exclusively studying WT. Additionally, for fluorescence correlation spectroscopy (FCS), we engineered a single cysteine at the amino terminus (Cys-Tau (2A)) for labeling via maleimide chemistry. These modified proteins have fibrillization and seeding properties similar to FL WT tau. To initiate fibrillization, we incubated 8 μM tau in 10 mM HEPES, 100 mM NaCl, and 8 μM heparin (1:1 ratio of FL tau to heparin) at 37°C for 72 hr without agitation. For cysteine labeling, we incubated 200 μL of 8 μM fibrils (monomer equivalent) and monomer with 0.025 mg of Alexa Fluor-488 (AF488) C5-maleimide (Invitrogen) and 80 μM Tetramethylrhodamine-5-maleimide (Sigma-Aldrich) overnight at 4°C with gentle rotation. We quenched excess dye with 10 mM DTT for 1 hr at room temperature. For limited heparin exposure, recombinant tau at 1 μM was incubated with heparin at 1 μM for 15 min, 1 hr and 4 hr at 37°C before purification of monomer via Superdex 200 column.

We employ the following terminology:

$M_i$: This refers to 'inert' tau monomer, whether recombinant or derived from control brain.

**$M_s$**: This refers to 'seed competent' monomer, whether derived from sonicated fibrils, heparin-treated monomer, or AD brain.

## Sonication and size exclusion chromatography (SEC)

We sonicated labeled and non-labeled fibrils using a Q700 Sonicator (QSonica) at a power of 100–110 watt (Amplitude 50) at 4°C for 3 hr. Samples were then centrifuged at 10,000 x g for 10 min and 1 mL of supernatant was loaded into a Superdex 200 Increase 10/300 GL column (GE Healthcare) and eluted in PBS buffer at 4°C. After measuring the protein content of each fraction with a Micro BCA assay (Thermo Scientific, Waltham MA) and/or fluorescence using a plate reader (Tecan M1000), we aliquoted and stored samples at −80°C or immediately used them in biochemical studies and cell seeding assays. Each aliquot was thawed immediately before use. The molecular weight/radius of proteins in each fraction was estimated by running gel filtration standards (Bio-Rad): Thyroglobulin (bovine) 670 kDa/8.5 nm; γ-globulin (bovine) 158 kDa/5.29 nm; Ovalbumin (chicken) 44 kDa/3.05 nm; myoglobin (horse) 17 kDa/2.04 nm; and vitamin $B_{12}$1.35 kDa/0.85 nm. In a prior publication (*Mirbaha et al., 2015*), *Figure 1E*, we demonstrated through use of crosslinking with SDS-PAGE that the SEC protocol used in this work reliably purifies monomer, dimer, and trimer.

## Size-cutoff filtration

Monomer, dimer and trimer fractions were passed through a 100 kDa MWCO filter (Corning) as instructed by the manufacturer (centrifuged at 15,000 x g for 15 min at 4°C). Filtered material was immediately collected and used in seeding assay along with the non-filtered samples of the same fraction at a final concentration of 100 nM, or analyzed by limited proteolysis. Protein concentration was determined before and after filtration by determining absorption at 205 nm.

## CD spectroscopy

Circular dichroism (CD) measurements were performed at 25°C on a Jasco J-815 spectropolarimeter using a 0.1 cm optical path length. 200 µL of 2 µM $_{Ms}$ or $M_i$ monomer was dialyzed onto 10 mM NaP and the spectra were measured at 0.10 nm intervals, with a band width of 1.0 nm, and scan speed of 10 nm/min. The spectrum represents the average of 4 scans in the range of 195 to 250 nm.

## Enzyme-linked immunosorbent assay

A total tau 'sandwich' ELISA was performed similarly to that described previously (*Acker et al., 2013*). Antibodies were kindly provided by Dr. Peter Davies (Albert Einstein College of Medicine). 96-well round-bottom plates (Corning) were coated for 48 hr at 4°C with DA-31 (aa 150–190) diluted in sodium bicarbonate buffer (6 µg/mL). Plates were rinsed with PBS three times, blocked for 2 hr at room temperature with Starting Block (Pierce), and rinsed with PBS five additional times. SEC fractions were diluted in SuperBlock solution (Pierce; 20% SuperBlock, diluted in TBS), and 50 µL sample was added per well. DA-9 (aa 102–150) was conjugated to HRP using the Lighting-Link HRP Conjugation Kit (Innova Biosciences), diluted 1:50 in SuperBlock solution, and 50 µL was added per well (15 µg/mL). Sample +detection antibody complexes were incubated overnight at 4°C. Plates were washed with PBS nine times with a 15 s incubation between each wash, and 75 µL 1-Step Ultra TMB Substrate Solution (Pierce) was added. Plates were developed for 30 min, and the reaction quenched with 2M sulfuric acid. Absorbance was measured at 450 nm using an Epoch plate reader (BioTek). Each plate contained a standard curve, and all samples were run in triplicate.

## Fluorescence correlation spectroscopy

FCS measurements were conducted on a Confocal/Multiphoton Zeiss LSM780 Inverted microscope (Carl Zeiss-Evotec, Jena, Germany), using a 40X water immersion objective as previously described (*Chattopadhyay et al., 2002*). Fluorescently labeled tau from SEC fractions (in PBS) was excited at 488 nm and 561 nm for 30 s, recording 10 times (*Buschmann et al., 2003*). The data analysis was performed with Origin 7.0 (OriginLab, Northampton, MA).

## Liposome-mediated transduction of tau seeds

Stable cell lines were plated at a density of 35,000 cells per well in a 96-well plate. After 18 hr, at 60% confluency, cells were transduced with protein seeds. Transduction complexes were made by

combining [8.75 μL Opti-MEM (Gibco) +1.25 μL Lipofectamine 2000 (Invitrogen)] with [Opti-MEM + proteopathic seeds] for a total volume of 20 μL per well. Liposome preparations were incubated at room temperature for 20 min before adding to cells. Cells were incubated with transduction complexes for 24 hr. Biosensor cells were confirmed as HEK293T by PowerPlex sequencing. Mycoplasma contamination was ruled out by PCR analysis using VenorGem (Sigma).

## FRET flow cytometry

Cells were harvested with 0.05% trypsin and fixed in 2% paraformaldehyde (Electron Microscopy Services) for 10 min, then resuspended in flow cytometry buffer. The MACSQuant VYB (Miltenyi) was used to perform FRET flow cytometry. To measure CFP and FRET, cells were excited with a 405 nm laser, and fluorescence was captured with 405/50 nm and 525/50 nm filters, respectively. To measure YFP, cells were excited with a 488 nm laser and fluorescence was captured with a 525/50 nm filter. To quantify FRET, we used a gating strategy similar to that previously described (*Holmes et al., 2014*). The integrated FRET density (IFD), defined as the percentage of FRET-positive cells multiplied by the median fluorescence intensity of FRET-positive cells, was used for all analyses. For each experiment,~20,000 cells were analyzed in triplicate. Analysis was performed using FlowJo v10 software (Treestar).

## Tau seeding in vitro

Recombinant full length (0N4R) tau monomer was purified as previously described (*Morozova et al., 2013*) at 1 mg/mL in BRB80 buffer (80 mM PIPES, 1 mM MgCl2, 1 mM EGTA, pH 6.8 with 0.3M NaCl) and boiled at 100°C for 5 min with 25 mM β-mercaptoethanol. The tau protein solution was then rapidly diluted 1:5 and cooled to 20°C in PBS, pH 7.4, to a final concentration of 0.2 mg/mL of tau and 5 mM β-mercaptoethanol. This solution was supplemented with Thioflavin T (ThT) to a final concentration of 20 μM and filtered through a sterile 0.2 μm filter. Reaction sizes of 195 μL were aliquoted from the prepared protein stock and thoroughly mixed with 5 μL of each sample at 100 nM monomer equivalent, or 5 μL of buffer control. For each sample, three different technical replicates were prepared. An opaque 96-well plate was prepared with a 3 mm glass bead added to each well to increase agitation. The recombinant tau solution was added to the plate in 200 μl reaction volumes. The plate was sealed with sealing tape to prevent evaporation and incubated in the plate reader (SpectraMax M2) at 37°C. ThT fluorescence was monitored over time with excitation and emission filters set to 444 nm and 485 nm, respectively. Fluorescence readings were taken every 5 min, with agitation for 5 s before each reading.

## Tau extraction from brain and characterization by SEC

0.5 g frontal lobe sections from AD patients at late Braak stage (VI) and age-matched controls lacking evident tau pathology were gently homogenized at 4°C in 5 mL of TBS buffer containing protease inhibitor cocktails (Roche) using a dounce homogenizer. Samples were centrifuged at 21,000 x g for 15 min at 4°C to remove cellular debris. Supernatant was partitioned into aliquots, snap frozen and stored at −80°C. Immunopurification was performed with HJ8.5 anti-tau antibody (*Yanamandra et al., 2013*) at a ratio of 1:50 (1 μg mAb per 50 μg of total protein), incubating overnight at 4°C while rotating. To each 1 mL of mAb/brain homogenate we added 200 μL of a 50% slurry protein G-agarose beads (Santa-Cruz). We washed the bead with TBS buffer before overnight incubation at 4°C. We then centrifuged the complexes at 1000 x g for 3 min and discarded the supernatant. Beads were washed with Ag/Ab Binding Buffer, pH 8.0 (Thermo Scientific) three times. Tau bound to the beads was eluted in 100 μL low pH elution buffer (Thermo Scientific), incubated at room temperature for 7 min, followed by neutralization with 10 μL Tris-base pH 8.5. This elution step was repeated once more with 50 μL elution buffer and 5 μL Tris-base pH 8.5 for a total of 165 μL. Samples were then centrifuged at 10,000 x g for 10 min, and the supernatant loaded onto a Superdex 200 Increase 10/300 GL column (GE Healthcare). SEC fractions were frozen at −80°C after evaluation of protein content by Micro BCA assay (Thermo Scientific).

To compare different extraction methods, fresh frozen frontal lobe section from an AD patient brain was suspended in TBS buffer containing protease inhibitor cocktails (Roche) at 10% w/vol in four portions. Samples were homogenized using three different devices: a dounce homogenizer, probe sonicator (Omni International), and tissue homogenizer (Power Gen 125, Fischer Scientific).

We also included one more condition of homogenizing with tissue homogenizer followed by probe sonication for 10 min. Samples were centrifuged at 21,000 x g for 15 min at 4°C to remove cellular debris. Supernatant was partitioned into aliquots followed by immunopurification.

To control for release of tau $M_s$ from fibrils in AD brain, a tau KO Mouse brain was divided into two halves, followed by spiking one half with recombinant fibrils and the other with fibril-derived $M_s$, both at final concentration of 10 µM monomer equivalent. Each was dounce homogenized, centrifuged, immunoprecipitated with HJ8.5 anti-tau antibody, and fractionated by SEC with identical techniques as used for human brain processing. SEC fractions were then used in seeding experiments.

## Analysis of heat denaturation data

We analyzed the IFD from measurements of temperature-dependent seeding using global fits to a proposed unimolecular heat denaturation reaction. This analysis rests on the Arrhenius equation *Laidler, 1984*:

$$k_U = Ae^{-\frac{E}{RT}}$$

where $k_U$ is the unfolding rate constant, $E$ is the activation energy, $R$ is the gas constant, $T$ is the temperature, and $A$ is the pre-exponential factor. For the unimodal model, the data were fit globally to:

$$\text{IFD}(t) = 100e^{-t/\tau}$$

Here, $t$ is the heat denaturation time and $\tau = 1/k_U$ is the unfolding time. A second, multimodal model was deployed to account for discrepancies in the early time points which appeared to suggest the presence of a lag phase in denaturation. In this model, the data were fit globally to

$$\text{IFD}(t) = 100; \qquad t \leq l_t$$
$$\text{IFD}(t) = 100e^{-(t-l_t)/\tau}; \quad t > l_t$$

where $l_t$ is the lag time given by

$$1/l_t = Be^{-\frac{E}{RT}}$$

and $B$ is a pre-exponential factor. We used the Akaike information criterion (AIC) to evaluate the best model as it quantifies the trade-off between goodness of fit and the complexity of the model (*von Bergen et al., 2001*). For least squares model fitting, AIC can be reduced to:

$$\text{AIC} = 2p + n\ln(\text{RSS}/n)$$

where $p$ is the number of parameters in the model, $n$ is the number of observations, and RSS is the residual sum of squares. The preferred model is the one with the minimum AIC. Here, we find AIC = 123 for the unimodal model and AIC = 105 for the multimodal model, which suggests the multimodal model is a better description of the denaturation data.

## Crosslinking, sample processing and LC-MS/MS analysis

$M_i$ and $M_s$ tau samples were prepared as described above. In all cases, tau preparations were crosslinked at a total protein concentration of ~0.1 mg/mL using 10–20 µg starting material. The crosslinking buffer was 50 mM HEPES-KOH (pH 7.4) containing 150 mM NaCl and 1 mM DTT. The crosslinking reaction was initiated by adding disuccinimidyl suberate (DSS) stock solution (25 mM DSS-$d_0$ and –$d_{12}$, Creative Molecules) in DMF to a final concentration of 1 mM. Samples were incubated at 37°C for 1 min. For the heparin-derived $M_s$ sample, heparin sulfate (Sigma) was added to a final concentration of 5 µM, followed by 1 mM DSS and the samples were incubated for 1 min at 37°C. Excess reagent was quenched by addition of ammonium hydrogen carbonate to 50 mM and incubation at 37°C for 30 min, and then flash frozen at −80°C. Absence of higher molecular species was confirmed by SDS-PAGE and coomassie stain. After the quenching step, samples were evaporated to dryness in a vacuum centrifuge and resuspended in 8M urea. Proteins were reduced with 2.5 mM TCEP (37°C, 30 min) and alkylated with 5 mM iodoacetamide (30 min, room temperature, protected from light). The sample solutions were diluted to 1M urea with 50 mM ammonium hydrogen carbonate and trypsin (Promega) was added at an enzyme-to-substrate ratio of 1:50. Proteolysis

was carried out at 37°C overnight followed by acidification with formic acid to 2% (v/v). Samples were then purified by solid-phase extraction using Sep-Pak tC18 cartridges (Waters) according to standard protocols. Samples were fractionated by size exclusion chromatography (SEC) on a Superdex Peptide column as described elsewhere (*Leitner et al., 2012*). Two fractions collected from SEC were evaporated to dryness and reconstituted in water/acetonitrile/formic acid (95:5:0.1, v/v/v) to a final concentration of approximately 0.5 µg/µl. 2 µL each were injected for duplicate LC-MS/MS analyses on an Eksigent 1D-NanoLC-Ultra HPLC system coupled to a Thermo Orbitrap Fusion Lumos Tribrid system (Thermo Scientific). Peptides were separated on self-packed New Objective PicoFrit columns (11 cm x 0.075 mm I.D.) containing Magic $C_{18}$ material (Michrom, 3 µm particle size, 200 Å pore size) at a flow rate of 300 nL/min using the following gradient. 0–5min = 5 %B, 5–95min = 5–35%B, 95–97 min = 35–95%B and 97–107min = 95 %B, where A = (water/acetonitrile/formic acid, 97:3:0.1) and B = (acetonitrile/water/formic acid, 97:3:0.1). The mass spectrometer was operated in data-dependent mode by selecting the five most abundant precursor ions (m/z 350–1600, charge state 3 + and above) from a preview scan and subjecting them to collision-induced dissociation (normalized collision energy = 35%, 30 ms activation). Fragment ions were detected at low resolution in the linear ion trap. Dynamic exclusion was enabled (repeat count 1, exclusion duration 30 s).

## Analysis of mass spectrometry data

Thermo. raw files were converted into the open. mzXML format using msconvert (proteowizard.sourceforge.net) and analyzed using an in-house version of xQuest (*Rinner et al., 2008*). Spectral pairs with a precursor mass difference of 12.075321 Da were extracted and searched against the respective FASTA databases containing Tau (TAU_HUMAN P10636-8).xQuest settings were as follows: Maximum number of missed cleavages (excluding the crosslinking site)=2, peptide length = 5–50 aa, fixed modifications = carbamidomethyl Cys (mass shift = 57.021460 Da), mass shift of the light crosslinker = 138.068080 Da, mass shift of mono-links = 156.078644 and 155.096428 Da, MS (*Chirita et al., 2005*) tolerance = 10 ppm, MS (*Kar et al., 2011*) tolerance = 0.2 Da for common ions and 0.3 Da for crosslink ions, search in ion-tag mode.For brain-derived samples we also included variable modifications including: Methionine oxidation = 15.99491, Ser/Thr/Tyr phosphorylation = 79.96633 and lysine ubiquitylation = 114.043 with nvariable_mod = 1. Post-search manual validation and filtering of the recombinant samples was performed using the following criteria: xQuest score >16, mass error between −4 and +7 ppm, %TIC >10, and a minimum peptide length of six aa. In addition, at least four assigned fragment ions (or at least three contiguous fragments) were required on each of the two peptides in a crosslink. False discovery rates (FDR) for the identified crosslinks were estimated using xprophet (*Rinner et al., 2008*). For the recombinant samples, $M_i$ and $M_s$, the FDR ranged from 6–10%. Post-search manual validation of the brain-derived samples was performed using the following criteria: xQuest score >7, mass error between −5 and +7 ppm, %TIC >10, and a minimum peptide length of six aa. In addition, at least four assigned fragment ions (or at least three contiguous fragments) were required on each of the two peptides in a crosslink. The FDRs for the brain samples were much higher and ranged between 20–25%. For triplicate datasets (N = 3) corresponding to the $M_i$ and $M_s$ boiling time course we computed consensus crosslink profiles enforcing that at least two of the three datasets contain a crosslink. Crosslink data were visualized using Xvis (*Grimm et al., 2015*). Average contact distance was computed by averaging the sequence separation between crosslink pairs in a given dataset. See *Figure 7—source datas 1* and *2*.

## Generation of structural models using XL-MS-derived constraints

High confidence crosslink pairs identified above were used to generate an ensemble of possible structures using a Rosetta protocol employing the crosslink pairs as structural restraints. The integration of XL-MS derived restraints have been previously used to refine structural models of large complexes (*Leitner et al., 2012*) and simpler heterodimeric complexes (*Kahraman et al., 2013*). Based on distance distributions of crosslink pairs mapped onto crystallographic structures we set a lower bound of 15 Å and an upper bound of 25 Å for lysine Cα pairs in our simulations. Importantly, in our simulations we weighted the constraint pairs as to allow some distances above the upper bound limit. The fragment library was supplanted by using chemical shifts derived from fibrillar tau ssNMR assignments (bmrb entry 17920) using csrosetta (*Lange et al., 2012*). We generated 1000 models

for each of the four XL-MS datasets on a high performance cluster (biohpc.swmed.edu). Representative structures were selected according to the low Rosetta score and radius of gyration. All plots were generated with gnuplot. All figures were generated using Pymol. See *Figure 9—source data 1*.

## Commandline used for *Ab Initio* protocol calculations with XL-MS restraints

AbinitioRelax.default.linuxgccrelease -in:file:fasta tau.fasta -file:frag3 tau.frags3.dat -file:frag9 tau.frags9.dat -nstruct 1000 -abinitio::increase_cycles 0.5 -abinitio::relax -score::weights score13_env_hb -abinitio::rg_reweight 0.5 -abinitio::rsd_wt_helix 0.5 -abinitio::rsd_wt_loop 0.5 -disable_co_filter true -out:file:silent csrosetta.out -constraints:cst_fa_file tau.cst -constraints:cst_file tau.cst -constraints:cst_weight 0.1 -constraints:cst_fa_weight 0.1 -loopfcst::coord_cst_weight 10.0

## Statistical analysis

Group mean values were analyzed by one-way ANOVA with Bonferroni post hoc significant differences test using GraphPad prism five software. Data in text and figures are represented as mean ± SD.

## Replicates and sample size

**Biological Replicates:** This refers to separately generated samples, e.g. analysis of samples from different individuals, in which the input represents a distinct biological source, or analysis of fibril preparations formed independently.

**Technical Replicates:** This refers to independently analyzed samples in which each would be expected otherwise to be identical, e.g. multiple wells in which the same cell line was treated with the same sample, or when a single brain is broken up into three fractions for identical analyses.

**Outliers:** We did not exclude outliers in any case.

**Inclusion/Exclusion of data:** In no case did we include or exclude data.

**Figure 1:** Biochemical purification of $M_i$ and $M_s$ and seeding into cells was carried out ~50 times over 3 years. In *Figure 1* the data shown is a representative dataset for $M_i$ and $M_s$ on SEC, seeding, microscopy and seeding following filtration. In vitro $M_s$ and $M_i$ aggregation was done once (using N = 3 independent technical replicates per fraction studied). Data from the titration of $M_i$ and $M_s$ seeding in cells is from a single representative experiment (from at least three independent studies), using technical triplicates for each sample. Primary data is available in *Figure 1—source data 1*.

**Figure 2**: CD experiments were carried out with three biological replicates. A single representative experiment is shown, and the data for $M_i$ and $M_s$ is an average of 4 technical replicates. The FCS experiment was repeated three times with biological replicates. A single representative scan is shown. Primary data is available in *Figure 2—source data 1*.

**Figure 3**: Seeding activity was measured in technical triplicates. Primary data is available in *Figure 3 source data 1*.

**Figure 4**: Heat denaturation was done in two biological replicates, for which a representative experiment is shown. Seeding activity was performed in technical triplicates. Primary data is available in *Figure 4—source data 1*.

**Figure 5**: Stability of monomers and assemblies in solution was tested in a single experiment, with 32 s analyses. Primary data is available in .

**Figure 6**: This single experiment is representative of at least three distinct biological replicates. Seeding activity was determined in technical triplicates (N = 3). Primary data is available in *Figure 6—source data 1*.

**Figure 6—figure supplement 1**: This experiment was performed in three biological replicates. A representative gel is shown.

**Figure 7**: Each XL-MS dataset was collected in biological triplicate across the different conditions ($M_i$: Recombinant; $M_s$: Fibril-derived; $M_s$: Heparin-derived). The figure indicates consensus crosslinks across triplicates for each condition. Primary data is available in *Figure 7—source data 1*, and *Figure 7—source data 2*.

**Figure 7—figure supplement 1**: This represents a histogram of consensus pairs across triplicates for each condition depicted in *Figure 7*. Primary data is available in *Figure 7—source data 1*, and *Figure 7—source data 2*.

**Figure 8**: SEC was performed at least three times, and a single representative experiment is shown. The seeding assay was performed in technical triplicate for each fraction. Spiking of tau KO brain with fibrils or $M_s$ was performed once, and the SEC was performed once, with the seeding assay performed in technical triplicate. XL-MS was performed in biological triplicate for Controls vs. AD. Primary data is available in *Figure 8—source datas 1* and *2*.

**Figure 8—figure supplement 1**: XL-MS data from different methods of homogenization was performed in four individual experiments on a single biological sample. Primary data is available primary data available in .

**Figure 9**: Rosetta simulations produced models from 1000 independent trajectories across four different conditions: $M_i$, $M_s$, Control, AD. The pymol session is available in *Figure 9—source data 1*.

**Figure 9—figure supplement 1**: Plot of the energetics of the ensemble of 1000 Rosetta models built for each condition. Primary data is available in *Figure 9—source data 2*.

**Figure 10**: Proteolysis experiments were carried out for $M_i$ and $M_s$ in technical triplicate for each time point. Primary data is available in *Figure 10—source data 1*.

**Figure 10—figure supplement 1**: Summary data from *Figure 10*, comparing peptide abundance between Mi and Ms is represented as technical triplicates for each data point presented. Primary data is available in *Figure 10 source data 2*.

## Kinetic analyses of $M_i$ and $M_s$ proteolysis

Limited proteolysis of $M_i$ and $M_s$ using trypsin was carried out in triplicate (N = 3) in 50 mM TEAB at 25°C. The enzyme to tau ratio was adjusted to 1:100 (wt/wt) with around 11 ug of $M_i$/$M_s$ present initially. The total reaction mixture volume was 60 µl. Aliquots were withdrawn from the reaction mixture at 1, 5, 15, 30, 60 and 120 min by using 10 µL of 10% trifluoroacetic acid (TFA) to quench the reaction (PH <3). The trypsin-digested peptides were then desalted using an Oasis HLB plate (Waters) and eluted with 100 µL 80% acetonitrile (ACN) containing 0.1% TFA. The solvent was evaporated in a SpeedVac concentrator and the dried samples were reconstituted in 20 µl of 2% acetonitrile, 0.1% TFA and 2 µl solution was used for by LC/MS/MS analysis, the analysis were performed on an Orbitrap Elite mass spectrometer (Thermo Electron) coupled to an Ultimate 3000 RSLC-Nano liquid chromatography systems (Dionex). Samples were injected onto a 75 µm i.d., 15 cm long Easy-Spray column (Thermo Scientific), and eluted with a gradient from 1 to 28% buffer B over 60 min. Buffer A contained 2% (v/v) ACN and 0.1% formic acid in water, and buffer B contained 80% (v/v) ACN, 10% (v/v) trifluoroethanol, and 0.1% formic acid in water. The mass spectrometer operated in positive ion mode with a source voltage of 2.8kV and an ion transfer tube temperature of 275°C. MS scans were acquired at 240,000 resolution in the Orbitrap and up to 14 MS/MS spectra were obtained in the ion trap for each full spectrum acquired using collision-induced dissociation (CID), with charge one ions rejected. Dynamic exclusion was set for 15 s after an ion was selected for fragmentation. Raw MS data files were searched against the appropriate protein database from Uniprot, and reversed decoy sequences appended (*Elias and Gygi, 2010*) by using Protein Discovery 2.2 (Thermo Fisher Scientific). Fragment and precursor tolerances of 20ppm and 0.6 Da were specified, and 12 missed cleavages were allowed. Carbamidomethylation of Cys was set as a fixed modification and oxidation of Met was set as a variable modification. Label-free quantitation of proteins across samples was performed. Average peptide intensity values were computed for all time points for each peptide across triplicates (N = 3). To estimate differences in kinetic profiles we calculated the median value of each profile and compared the $M_i$ to $M_s$ ratio. See *Figure 10—source datas 1* and *2*.

## Acknowledgements

We thank Peter Davies for generously providing antibody reagents and ELISA protocol guidance. This work was supported by grants from the Tau Consortium and NIH grants awarded to 1R01NS071835 (MID), R01NS089932 (RVP and MID), and the Effie Marie Cain Endowed Scholarship (LAJ). We appreciate the help of the Live Cell Imaging Core Facility administered by Katherine Luby-Phelps, Ph.D., and the Proteomics Core Facility at the University of Texas Southwestern Medical Center. We thank Hilal Lashuel, Ph.D. for critical comments on the manuscript.

## Additional information

### Funding

| Funder | Grant reference number | Author |
| --- | --- | --- |
| National Institutes of Health | R01NS089932 | Rohit V Pappu<br>Marc I Diamond |
| Effie Marie Cain Endowed Scholarship | | Lukasz A Joachimiak |
| National Institutes of Health | R01AG048678 | Marc I Diamond |
| Rainwater Charitable Foundation | | Marc I Diamond |

The funders had no role in study design, data collection and interpretation, or the decision to submit the work for publication.

### Author contributions

Hilda Mirbaha, Conceptualization, Resources, Supervision, Writing—original draft, Writing—review and editing; Dailu Chen, Conceptualization, Data curation, Formal analysis, Supervision, Investigation, Methodology, Writing—original draft; Olga A Morazova, Kiersten M Ruff, Apurwa M Sharma, Xiaohua Liu, Formal analysis, Investigation, Methodology; Mohammad Goodarzi, Data curation, Formal analysis, Investigation, Methodology; Rohit V Pappu, Investigation, Methodology, Writing—original draft; David W Colby, Resources, Formal analysis, Investigation, Methodology; Hamid Mirzaei, Resources, Data curation, Formal analysis, Supervision, Investigation, Methodology, Writing—original draft, Project administration; Lukasz A Joachimiak, Data curation, Formal analysis, Supervision, Methodology, Writing—original draft, Project administration, Writing—review and editing; Marc I Diamond, Formal Analysis

### Author ORCIDs

Kiersten M Ruff (iD) http://orcid.org/0000-0003-3240-1856
Rohit V Pappu (iD) https://orcid.org/0000-0003-2568-1378
Hamid Mirzaei (iD) http://orcid.org/0000-0001-5013-3432
Lukasz A Joachimiak (iD) http://orcid.org/0000-0003-3061-5850
Marc I Diamond (iD) http://orcid.org/0000-0002-8085-7770

### Decision letter and Author response

Decision letter https://doi.org/10.7554/eLife.36584.034
Author response https://doi.org/10.7554/eLife.36584.035

## Additional files

### Supplementary files

• Transparent reporting form
DOI: https://doi.org/10.7554/eLife.36584.032

### Data availability

All data generated or analysed during this study are included in the manuscript and supporting files.

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
