## [Decision Letter]

[Editors’ note: a previous version of this study was rejected after peer review, but the authors submitted for reconsideration. The first decision letter after peer review is shown below.]

Thank you for submitting your work entitled "Inert and seed-competent tau monomers elucidate the structural origins of aggregation" for consideration by *eLife*. Your article has been reviewed by three peer reviewers, and the evaluation has been overseen by a Senior/Reviewing Editor. The reviewers have opted to remain anonymous.

Our decision has been reached after consultation between the reviewers. Based on these discussions and the individual reviews below, we regret to inform you that your work will not be considered further for publication in *eLife*.

The reviewers raised major concerns over the principal finding that a form of heat-stable tau monomer possesses seeding activity. In this case, it seems that generating adequate controls could take longer than 2-3 months, and the outcome of such experiments seems uncertain. Furthermore, the secondary finding of constraints for monomer seed structure also requires major revisions or might even need to be removed as pointed out by reviewer 1. For these reasons, we cannot proceed with your paper. However, if you were able to gather the data needed to fully support your hypothesis, we will be prepared to give this study another round of consideration and to send it to the same reviewers.

Reviewer #1:

1) In this paper, the authors claim to have identified two distinct monomeric forms of tau, where one form is seed-competent and the other is inert. Somehow, the seed-competent monomer can be heated to 95°C for 3 hours and still be conformationally distinct from the inert monomer. This is an extraordinary claim that will need extraordinary evidence to support it. What kind of intra-molecular interactions would resist this type of treatment? Especially, given that tau is disordered to start with. What if there were contaminating filamentous tau species (for example, formed from self-assembly due to prolonged storage and/or freeze-thaw)? Such filaments may have a far greater seed-competency than the monomer, dimer and trimer fractions tested and, therefore, even a very small amount of contamination could have a big effect. This might form a much simpler explanation of the presented results. For this reason, it would be important to know how the seeding activities of the M_s_ fraction and AD-derived tau monomer compared to the total pool of molecular tau species and to tau filaments alone. The majority of published work on seeded tau aggregation has been conducted using filamentous tau seeds and it would, therefore, be very useful to have this direct comparison presented here. At present, the authors only test the effects of contamination by dimers and trimers of tau.

2) Also, the in vitro experiments all start with heparin-induced filament formation. These are then sonicated, centrifuged and put over a SEC column to obtain the seed-competent monomer. Yet, the authors in multiple places (including the title) state that they have elucidated the structural 'origins' of tau aggregation. This claim is not supported by the data: the data work backwards from filaments to monomers, which is not necessarily the same as starting from some 'aggregation origin' to form filaments. In this context, the authors' claim that the use of a dounce homogenizer would prevent the liberation of tau monomers from filaments during brain homogenization is also important. I am not convinced this is actually the case. Neurofibrillary tangles are very large, filling most of the somato-dendritic compartment, and are likely broken up by a dounce homogenizer. The authors should directly test this before concluding that AD brain-derived seed-competent tau monomers are not actually derived from tau filaments. The centrifugation steps used prior to size exclusion chromatography (10,000 xg, 10 min, for recombinant protein and 21,000 xg, 15 min, for brain homogenates) would pellet the majority of filamentous tau aggregates and tangles, thereby excluding them from comparisons.

3) The authors present cross-link mass-spectrometry data to study the conformation of the seed-competent and inert monomers. These experiments lead to the identification of a few distance restraints between different regions of the protein (Figure 7), which are then used to model the conformations using ROSETTA (Figure 8). It is outrageous to assume one can get anywhere near reliable 3D atomic models from such little data (the authors even describe differences in the accessibility of 2 hexapeptides inside this ~400-residue long protein!). The XL-MS data may be valuable, but Figure 8 and the corresponding section "Models of…" in the Results should be removed in a revised version of the paper. One prediction from Figure 4 would be that the red cross-link in Figure 7D and F would persist if samples were first incubated for 3 hours at 95^o^C. In the context of necessary extraordinary evidence, the authors should include this experiment in a revised version of the manuscript.

Reviewer #2:

This paper provides evidence for a provocative hypothesis, that tau monomer with the VQIINK and VQIVYK sequences exposed in a consistent conformational ensemble is one of the seeds for initiating Tau aggregation.

While the data in this paper are generally supportive of this hypothesis, this paper was very hard to read/review critically as the authors forced the reviewers to go back to already published papers to get all the details-this paper has to be self-contained if I am to review the revised version.

Since M_s_ self-assembles efficiently under conditions where seeding is being assessed in the cell reporter line (heparin and Lipofectamine), how can the authors be confident that the monomer is doing the seeding and not a self-assembled M_s_? This needs to be explicitly addressed and the conditions for the cell reported seeding experiments need to be discussed in detail.

Has the heparin + Lipofectamine only control been done in the cell reporter line? I am sorry if I missed this.

It seems important to have heavy and light Tau 50:50 in the crosslinking experiment to be certain that oligomeric crosslinks are not contributing.

This paper seems to be an important contribution, and I hope the authors can revise the paper not assuming the readers have read all their papers, while integrating the control results or discussing them if they have been done.

Reviewer #3:

This article characterizes the structural requirements for tau seeding activity. The approach is innovative, in that the identification of seeding activity follows physical characterization rather than kinetic analysis. The results are novel, in that they identify a monomeric form of full-length tau as having seeding activity. The work also is relevant for tauopathic disease, as it integrates analysis of brain tissue derived tau in the analysis. Despite these strengths, a weakness of the work is that it relies solely on hydrodynamic analysis to deduce tau structure, which depends on molecular shape as well as size. As a result, conclusions regarding the size of tau seeds seem premature and incomplete. Specifically, there is risk that the monomeric seeding activity identified by authors reflects break down of tau into aggregation-prone fragments.

1) Figure 1 shows size exclusion chromatograms for globular protein standards and for various tau species. Leaving aside for a moment that SEC should be calibrated in terms of hydrodynamic radius rather than MW, it is worrying that M_i_ tau elutes with ovalbumin, a globular protein of similar MW but a much smaller hydrodynamic radius (half!) than 2N4R tau protein (PMID: 14769047, 15823045, 19226187, et al.). Since M_i_ is a gel-filtered form of M_r_, this result could be a sign of proteolysis, which can generate small monomeric fragments having greater aggregation propensity and possibly seeding activity. Proteolysis is always a concern when working with tau protein, but especially for M_s_ because it is generated by extensively incubation/sonication at elevated temperatures. Authors should provide SDS-PAGE or M_S_ data on M_h_, M_i_, M_r_, and M_s_ so as to disclose to what extent these preparations are proteolyzed. Because the seeding assays are sensitive, generation of even minor amounts of aggregation-competent fragments could yield a positive and confounding seeding result.

2) Since SEC is sensitive to shape as well as size, it is inappropriate to interpret eluted tau species only in terms of size. FCS depends on shape and size too, so it does not provide a "second opinion". The seed depletion and spiking experiments (Figure 3) don't control for this problem either.

3) It seems that M_r_ should be treated the same as M_s_, i.e. incubated and then sonicated, if it is to yield an M_i_ that can act as a control for M_s_.

4) The legend to Figure 1D states that "Seeding activity of each fraction was evaluated by SEC". The meaning of this statement is not clear.

5) The seeding assay used 0N4R tau, whereas M_i_ and M_s_ seeds were prepared from 2N4R tau. Why was this done?

6) In the recently reported structure of tau fibrils (PMID: 28678775), the segment VQIVYK interacts with 373THKLTF378. Can authors comment on solvent exposure of THKLTF in their models?

[Editors’ note: what now follows is the decision letter after the authors submitted for further consideration.]

Thank you for resubmitting your work entitled "Inert and seed-competent tau monomers suggest structural origins of aggregation" for further consideration at *eLife*. Your revised article has been favorably evaluated by Anna Akhmanova as the Senior/Reviewing Editor, and three reviewers.

The major issues raised by the reviewers have been addressed in revision. Although there was some remaining skepticism about monomeric Tau as the seed, this paper merits publication. Undoubtedly, this hypothesis will generate a lot of interest and catalyze further experimentation to scrutinize this concept. This is good for the field, because if this paper ultimately stands the test of time, it will change how we think about tauopathies.

---

## [Author Response]

[Editors’ note: the author responses to the first round of peer review follow.]

Reviewer #1:1) In this paper, the authors claim to have identified two distinct monomeric forms of tau, where one form is seed-competent and the other is inert. Somehow, the seed-competent monomer can be heated to 95°C for 3 hours and still be conformationally distinct from the inert monomer. This is an extraordinary claim that will need extraordinary evidence to support it. What kind of intra-molecular interactions would resist this type of treatment? Especially, given that tau is disordered to start with.

We were indeed initially as surprised as the reviewer to see the results of these tests, which imply strong, but not unheard-of, protein stability. Indeed, prion infectivity is well known to be resistant to boiling and autoclaving. We thus used XL-MS to test for structural changes over time at 95°C. In Figure 7 we have now studied two forms of seed-competent monomer (fibril-derived, and heparin-derived), tracking the persistence of the critical crosslinks over time. We observe that this crosslink tracks with seeding activity, being detectable at 3h, but gone by 24h of heating at 95°C. We also note (in unpublished data) that 8M urea treatment completely abolishes secondary structure detected by XL-MS, as would be predicted.

One of the major points of this paper is to challenge the notion that tau being “disordered” is synonymous with “unstructured.” Our data are consistent with it being a protein that adopts multiple, stable conformational ensembles, even as a monomer in solution. In support of this general notion, we were excited to read a recent publication from the Tanaka laboratory which reports that, as we have observed for tau, Sup35 monomer (another “intrinsically disordered” prion protein) exhibits signs of discrete local structures that trigger it to form distinct strains (Ohhashi et al., 2018).

What if there were contaminating filamentous tau species (for example, formed from self-assembly due to prolonged storage and/or freeze-thaw)? Such filaments may have a far greater seed-competency than the monomer, dimer and trimer fractions tested and, therefore, even a very small amount of contamination could have a big effect. This might form a much simpler explanation of the presented results.

This is an excellent hypothesis that we have excluded by multiple controls:

a) In all cases except those specified to involve a freeze-thaw, we immediately took monomer preps from the SEC column to transduce cells. There was no prolonged storage of frozen material.

b) In Figure 1 we now pass putative monomer through a 100kD filter prior to seeding, and find that the activity passes efficiently through the filter. By contrast, dimer is largely blocked, and trimer is almost completely blocked.

c) In Figures 1 and 6, we directly compare seeding activity of monomer, dimer, trimer, and fibrillar species in dose responses. Seeding activity per nM of material is actually very similar across assembly sizes, which argues for a proportional increase in seeding activity for fibrils per molecule (i.e. assemblies are more efficient than monomer). But for dimers and trimers (most likely to contaminate the SEC preps) to see artifactual seeding activity would require enormous contamination.

d) In Figure 3, within the limits of detection, the SEC column perfectly excludes larger species from the monomer fraction.

e)The SEC column is loaded with forms of tau that are soluble after a 10,000 x g spin, excluding large fibrils.

For this reason, it would be important to know how the seeding activities of the M_s_ fraction and AD-derived tau monomer compared to the total pool of molecular tau species and to tau filaments alone.

The seeding activity of recombinant monomer and fibrils are remarkably similar. Please see experiments cited in Figure 6. We did not directly compare recombinant M_s_ prepared in vitro with AD-derived M_s_, however we have noted that they have relatively similar seeding efficiencies in multiple experiments.

The majority of published work on seeded tau aggregation has been conducted using filamentous tau seeds and it would, therefore, be very useful to have this direct comparison presented here. At present, the authors only test the effects of contamination by dimers and trimers of tau.

Based on our second response to reviewer #1, we contend that it is essentially impossible that filamentous tau is significantly contaminating our preparations. In Figure 6C we have now directly compared the seeding efficiency between monomer and fibrils (sonicated and unsonicated). We also refer the reviewer to Figure 8D. In this experiment we spiked tau KO mouse brain with recombinant fibrils or M_s_, and carried out our standard purification with SEC. In this case, we isolated minimal signal from fibril-spiked brain in the ~20-mer fraction, but no signal in the monomer fraction. By contrast, after spiking the brain preparation with M_s_ we recovered robust monomer signal. It is thus highly unlikely that fibrils are contaminating our preparations.

2) Also, the in vitro experiments all start with heparin-induced filament formation. These are then sonicated, centrifuged and put over a SEC column to obtain the seed-competent monomer. Yet, the authors in multiple places (including the title) state that they have elucidated the structural 'origins' of tau aggregation. This claim is not supported by the data: the data work backwards from filaments to monomers, which is not necessarily the same as starting from some 'aggregation origin' to form filaments.

The reviewer is correct to point out that inferring “origins” of aggregation solely based on species fragmented from fibrils would be fraught. However, in addition to this approach (which is only how we first approached the question), we have also studied tau monomer that is converted by heparin exposure from an inert to seed-competent form, and have also isolated and studied brain-derived tau monomer. In Figure 6A we use brief exposure to heparin (prior to any larger assemblies forming) to create seed-competent monomer. Please note in Figure 6A that at 15 minutes the seeding activity purifies predominantly in the monomer fraction, and not larger assemblies. Thus, there is essentially no chance that the heparin-induced monomer we isolate was previously part of a larger assembly. Second, in Figure 8 we purify tau monomer from AD brain (controlling for possible liberation of tau monomer from fibrils (Figure 8D). This AD-derived monomer (but not control brain monomer) has intrinsic properties of self-assembly and seeding. Since (in the absence of heparin) we do not have “catalytic” control of tau monomer’s conformational change in vitro, to the best of our ability we have attempted to isolate a seed-competent monomer that was not previously part of a fibril. We thus feel justified in referring to structural “origins” of aggregation as we study this monomer in comparison to an inert form. However, we defer to the editor with regard to this disagreement in terminology.

In this context, the authors' claim that the use of a dounce homogenizer would prevent the liberation of tau monomers from filaments during brain homogenization is also important. I am not convinced this is actually the case.

We took very seriously this critique, and thus included a new control, described in Figure 8D of spiking mouse tau KO brain either with fibrils or M_s_ prior to dounce homogenization. There is no detectable liberation of tau monomer from fibrils following dounce homogenization.

Neurofibrillary tangles are very large, filling most of the somato-dendritic compartment, and are likely broken up by a dounce homogenizer. The authors should directly test this before concluding that AD brain-derived seed-competent tau monomers are not actually derived from tau filaments. The centrifugation steps used prior to size exclusion chromatography (10,000 xg, 10 min, for recombinant protein and 21,000 xg, 15 min, for brain homogenates) would pellet the majority of filamentous tau aggregates and tangles, thereby excluding them from comparisons.

We agree and are perhaps misunderstanding the point of this comment. As we are attempting here to define the smallest unit of seeding activity, this seems to be an appropriate step in the preparation of material.

3) The authors present cross-link mass-spectrometry data to study the conformation of the seed-competent and inert monomers. These experiments lead to the identification of a few distance restraints between different regions of the protein (Figure 7), which are then used to model the conformations using ROSETTA (Figure 8). It is outrageous to assume one can get anywhere near reliable 3D atomic models from such little data (the authors even describe differences in the accessibility of 2 hexapeptides inside this ~400-residue long protein!).

First, we thank the reviewer for pointing out a certain absurdity in attributing any structure to the extreme N- and C-termini of tau. Although we never attempted to imply 3D structural knowledge of tau based on XL-MS, we have now studiously avoided any reference to “structure” as it would be implied by NMR, cryo-EM, or crystallography. Instead, we have attempted to make very clear that XL-MS provides restraints for a “structural model.” Furthermore, we have now conscientiously avoided any reference to global tau structure, and have focused only on putative hairpin regions in the repeat domain. Based on the fragment libraries employed in our ROSETTA simulation there is a propensity to form secondary structure surrounding the two amyloid cores, VQIVYK and VQIINK, which leads to a diversity of models that contain hairpin-like structures within the RD1/RD2 or RD2/DR3 interface. When the seed-competent crosslinks to the RD2 region are used in the simulation we shift the equilibrium away from hairpin conformations at these sites, more so in RD1/RD2 than in RD2/RD3. Furthermore, we have limited any interpretation of XL-MS to regions in which multiple consensus crosslinks to RD2 exist. This allows us to build a model that is in fact fairly testable (and which we are pursuing in ongoing work). To this end, in Figure 10 we have now added limited proteolysis data that is consistent with our model. We hope we have made clear in the Results and Discussion that actual atomic-level resolution will require different types of studies. XL-MS and the model it supports provide a starting point with which to investigate actual structures in tau that allow it to self-assemble and act as a seed. In our reference to a “structural model” we are only intending to leave the reader with an intriguing and plausible hypothesis for our findings.

The XL-MS data may be valuable, but Figure 8 and the corresponding section "Models of…" in the Results should be removed in a revised version of the paper. One prediction from Figure 4 would be that the red cross-link in Figure 7D and F would persist if samples were first incubated for 3 hours at 95^o^C. In the context of necessary extraordinary evidence, the authors should include this experiment in a revised version of the manuscript.

Based on this excellent suggestion we have now included this data in a new Figure 7. We hope the reviewer is satisfied with the qualifications we have put on the interpretations of our XL-MS results, and don’t believe that proposing a very limited, local structural model based on ROSETTA is too bold.

Reviewer #2:This paper provides evidence for a provocative hypothesis, that tau monomer with the VQIINK and VQIVYK sequences exposed in a consistent conformational ensemble is one of the seeds for initiating Tau aggregation.While the data in this paper are generally supportive of this hypothesis, this paper was very hard to read/review critically as the authors forced the reviewers to go back to already published papers to get all the details-this paper has to be self-contained if I am to review the revised version.

We have attempted to elucidate all methods used in this version.

Since M_s_ self-assembles efficiently under conditions where seeding is being assessed in the cell reporter line (heparin and Lipofectamine), how can the authors be confident that the monomer is doing the seeding and not a self-assembled M_s_? This needs to be explicitly addressed and the conditions for the cell reported seeding experiments need to be discussed in detail.

This is indeed a difficult question. The reviewer correctly points out that it is impossible to know what exactly happens within cells. We believe the data in Figure 5 and 8, however, are important. In these cases, we observe that monomer assembles quickly to form larger seed-competent assemblies (especially in the case of AD-derived monomer), while the larger assemblies themselves (dimer, trimer) will not self-associate. We are not certain how further to discuss the cell seeding experiments, other than to describe our methods, which we hope now are quite clear, and have been extensively reported in prior publications.

Has the heparin + Lipofectamine only control been done in the cell reporter line? I am sorry if I missed this.

Every experiment includes Lipofectamine-treated buffer alone as a control. In experiments summarized in Figure 6 we also “transfected” cells with heparin alone and measured seeding. There was no effect, and we have alluded to this in the figure legend. We didn’t see the need to include completely negative data for this, but can include it if desired. Please note also that we see seeding activity from AD-derived monomer that has never had exposure to heparin.

It seems important to have heavy and light Tau 50:50 in the crosslinking experiment to be certain that oligomeric crosslinks are not contributing.

In the XL-MS experiments we used multiple controls to rule out multimers. Please note Figure 6—figure supplement 1, in which we display an example of a gel in which we have studied cross-linked heparin-induced M_s_, and find no evidence of multimerization. Critically, as part of our experimental protocol in every case when crosslinking monomer we run a control gel to rule out higher-order species (dimers, etc.). Additionally, when we probed the structures of M_i_ and M_s_ using limited proteolysis we passed the two forms of tau through a 100kD filter immediately prior to trypsin exposure, thus excluding multimers.

Reviewer #3:[…] A weakness of the work is that it relies solely on hydrodynamic analysis to deduce tau structure, which depends on molecular shape as well as size. As a result, conclusions regarding the size of tau seeds seem premature and incomplete. Specifically, there is risk that the monomeric seeding activity identified by authors reflects break down of tau into aggregation-prone fragments.

In this manuscript we have indeed used hydrodynamic analyses to infer sizes of assemblies, resolving those that are monomer from dimers, trimers, etc. We make no claims about the precise hydrodynamic radii. Additionally, in the current resubmission, we use other methods to infer monomer structure (e.g. double-label fluorescence correlation spectroscopy, 100kD filter cutoffs, and molecular crosslinking to confirm monomer structure). In responses below we address the question of seed-competent tau fragments in more detail.

1) Figure 1 shows size exclusion chromatograms for globular protein standards and for various tau species. Leaving aside for a moment that SEC should be calibrated in terms of hydrodynamic radius rather than MW, it is worrying that M_i_ tau elutes with ovalbumin, a globular protein of similar MW but a much smaller hydrodynamic radius (half!) than 2N4R tau protein (PMID: 14769047, 15823045, 19226187, et al.).

We thank the reviewer for educating us as to an appropriate indicator for standards on SEC, and have altered our figures accordingly. We do not intend to use SEC on the Superdex 200 column as a precise marker of hydrodynamic radius or molecular size. Instead we are simply seeking to pick fractions where tau monomer is likely to be separated from larger assemblies. In choosing the fraction B5 as the source of monomer, we do so recognizing that tau monomer appears to reflect an ensemble of structures that are slightly polydisperse as they come off of the column (especially in comparison to ovalbumin, which is more monodisperse). In multiple other experiments described especially in response to reviewer #1, we have attempted to test whether the minimal tau seed could be a multimer, and we find no evidence for this. As a final point, in our prior manuscript (Mirbaha et al., 2015), we carried out crosslinking of assemblies derived from putative monomer, dimer, and trimer fractions, and ran out the samples on a gel (Figure 1E, F, G). While we were studying tau RD, please note the fidelity of our SEC, in which we observed crosslinked assemblies that correlated with our predicted assembly sizes.

Since M_i_ is a gel-filtered form of M_r_, this result could be a sign of proteolysis, which can generate small monomeric fragments having greater aggregation propensity and possibly seeding activity. Proteolysis is always a concern when working with tau protein, but especially for M_s_ because it is generated by extensively incubation/sonication at elevated temperatures.

While we have been concerned about proteolysis, we have found no evidence of this in the absence of sonication (which indeed can produce fragments, as in Figure 6—figure supplement 1A). First of all, we emphasize that we find seed-competent tau monomer in conditions that do not involve sonication, and which do not change tau size. For example, in the new manuscript, please see Figure 6—figure supplement 1B, in which we have taken inert and heparin-induced seed-competent monomers, crosslinked them (or not) and resolved them by SDS-PAGE. While there are several species visible (it is very difficult to have a completely pure monomer preparation), even with the over-loaded gel the reviewer will observe that there is no evidence of significantly smaller fragments or larger assemblies associated with the seeding activity. We have also carried out multiple controls to rule out multimerization, even of small fragments, as referenced in prior responses. In particular, we refer the reviewer to double-label FCS experiments in Figure 2, in which we find no evidence that multiple labels are incorporated into putative monomers, while we easily detect them in putative dimers and trimers. Furthermore, the heat denaturation steps, which we show are sufficient to dissociate oligomers, do not change the seeding activity (or elution characteristics) of the seed-competent monomer. Finally, in Figure 7C we observe that sonication has no effect on the seeding activity of otherwise inert monomer, M_i_, despite causing a minor degree of fragmentation (Figure 6—figure supplement 1A). We believe at this point the data do not support the hypothesis that we are somehow purifying tiny amounts of seed-competent, aggregated fragments that account for the activity of M_s_.

Authors should provide SDS-PAGE or M_S_ data on M_h_, M_i_, M_r_, and M_s_ so as to disclose to what extent these preparations are proteolyzed. Because the seeding assays are sensitive, generation of even minor amounts of aggregation-competent fragments could yield a positive and confounding seeding result.

This is an excellent caveat. In response to this reviewer we performed SDS-

PAGE on sonicated monomer and observed evidence of fragmentation (Figure 6—figure supplement 1A). In Figure 6C we observe that sonication does not produce seed-competent species from M_i_. Nonetheless we additionally confirmed by other means that we were not generating aggregation-prone fragments by simply treating tau monomer (confirmed intact) with heparin, and also by isolating seed-competent monomer through gentle methods from AD brain. It seems to us to be incredibly unlikely that 15min of heparin exposure is sufficient to fragment tau to form cryptic seed-competent fragments than then spontaneously self-assemble.

2) Since SEC is sensitive to shape as well as size, it is inappropriate to interpret eluted tau species only in terms of size. FCS depends on shape and size too, so it does not provide a "second opinion". The seed depletion and spiking experiments (Figure 3) don't control for this problem either.

We take the reviewer’s point, and would emphasize only that control proteins elute from the SEC as rough markers of tau size. We make no specific reference to the “molecular weight” of the tau assemblies we elute from the SEC columns. Furthermore, the column we used (Superdex 200) is designed to resolve an enormous window of assembly sizes. It is not ideal to resolve ovalbumin and tau, and thus it is not surprising that these two proteins co-elute to some extent. We have predominantly been interested in resolving monomer, dimer, trimer, etc., and not attempting to determine a precise hydrodynamic radius. While FCS and SEC do use similar metrics to infer size, they are not identical (FCS does not involve interaction of protein with a column substrate), and the use of double-label FCS enables us to rule out multimeric assemblies of tau within the putative monomer fraction. Likewise, the seed depletion and spiking experiments are not done to determine size, per se, but to rule out the possibility of contamination of larger species (putative dimers and trimers) into the monomer fraction.

3) It seems that M_r_ should be treated the same as M_s_, i.e. incubated and then sonicated, if it is to yield an M_i_ that can act as a control for M_s_.

As part of our revision we have directly tested whether sonication of M_i_ produces seeding activity in the absence of heparin (Figure 6C). It does not. We would also point out that we can produce M_s_ without sonication, whether by brief (15min) exposure of recombinant monomer to heparin, or by purification of monomer from AD brain.

4) The legend to Figure 1D states that "Seeding activity of each fraction was evaluated by SEC". The meaning of this statement is not clear.

We have clarified this statement.

5) The seeding assay used 0N4R tau, whereas M_i_ and M_s_ seeds were prepared from 2N4R tau. Why was this done?

This was done purely for “historical” reasons because the Colby lab uses this form of tau for their seeding assays.

6) In the recently reported structure of tau fibrils (PMID: 28678775), the segment VQIVYK interacts with 373THKLTF378. Can authors comment on solvent exposure of THKLTF in their models?

The R4/R’ interface (which includes the THKLTF region) according to secondary structure and modeling analysis as well as previous work is predicted to be disordered. In our models, both M_i_ and M_s_, the THKLTF region is solvent exposed and not engaged in any interactions. The published cryoEM structure of the PHFs is also not consistent with many of the previous studies using solid state NMR and other methods. The conditions under which the fibrils form and the presence of the PTMs could favor a specific conformational arrangement of a fibril. Notably, the core of the fibril mediated by RD3 and RD4 in cryoEM structure buries many charged residues, which is quite surprising as this should destabilize the contacts between VQIVYK and THKLTF. More experiments will reveal the importance of this regions role in fibrils, in particular in strains, beyond this structure of a fibril derived from a single AD patient as recently reported.